# REVEALING AND MITIGATING OVER-ATTENTION IN KNOWLEDGE EDITING

**Pinzheng Wang**[1,2]   **Zecheng Tang**[1,2]   **Keyan Zhou**[1,2]
**Juntao Li**[1,2]*   **Qiaoming Zhu**[1,2]   **Min Zhang**[1,2]
[1]School of Computer Science and Technology, Soochow University
[2]Key Laboratory of Data Intelligence and Advanced Computing, Soochow University
`{pzwang1,zctang,kyzhou}@stu.sud.edu.cn`
`{ljt,qmzhu,minzhang}@suda.edu.cn`

## ABSTRACT

Large Language Models (LLMs) have demonstrated superior performance across a wide range of tasks, but they still exhibit undesirable errors due to incorrect knowledge learned from the training data. To avoid this, knowledge editing methods emerged to precisely edit the specific model knowledge via efficiently modifying a very small percentage of parameters. However, those methods can lead to the problem of **Specificity Failure**, where the existing knowledge and capabilities are severely degraded due to editing. Our preliminary indicates that Specificity Failure primarily stems from the model's attention heads assigning excessive attention scores to entities related to the edited knowledge, thereby unduly focusing on specific snippets within the context, which we denote as the **Attention Drift** phenomenon. To mitigate such Attention Drift issue, we introduce a simple yet effective method *Selective Attention Drift Restriction* (**SADR**), which introduces an additional regularization term during the knowledge editing process to restrict changes in the attention weight distribution, thereby preventing undue focus on the edited entity. Experiments on five frequently used strong LLMs demonstrate the effectiveness of our method, where SADR can significantly mitigate Specificity Failure in the predominant knowledge editing tasks[1].

## 1 INTRODUCTION

Large language models (LLMs) have demonstrated outstanding performance on various downstream natural language processing tasks, e.g., dialogue generation (Ni et al., 2023; Yang et al., 2024a), attributed to the numerous inherent knowledge (Roberts et al., 2020; Cao et al., 2023). However, many unpredictable errors inevitably arise due to the model's inner defect, which stems from negative or outdated samples within the extensive pre-training datasets (Balachandran et al., 2022). These imperfections can lead to the propagation of misinformation or biased outputs (Li et al., 2023a; Tang et al., 2023), undermining the reliability and robustness of the models in real-world applications.

One straightforward way to mitigate such an issue is to modify the knowledge of LLMs by directly fine-tuning the model with specific data. However, such a direct training method is uncontrollable and carries a significant risk of over-fitting (Kirkpatrick et al., 2017; Zhu et al., 2020). Consequently, knowledge editing methods (Meng et al., 2022; 2023) aim to efficiently modify specific knowledge within a limited subset of parameters while theoretically ensuring that the rest of the model's knowledge remains unchanged. However, when employing the knowledge editing methods, the instability feature limits their potential (Hoelscher-Obermaier et al., 2023). Specifically, we find that when content related to the edited knowledge appears in the context, it can inadvertently corrupt pre-existing knowledge, which we define as **Specificity Failure** in this paper. For instance, a language model, edited with a new knowledge—*"Eiffel Towel" is in "New York"* rather than *"Eiffel Towel" is in "Paris"* (Figure 1(a) and (b)), tends to predict *"Pyramids" is in "New York"* (Figure 1(c)), which is inconsistent with the original knowledge embedded in the model before editing, i.e., *"Pyramids" is*

---

*Corresponding author
[1]Code, dataset and an interactive demo notebook: https://github.com/PinzhengWang322/Reveal_Attention_Drift.

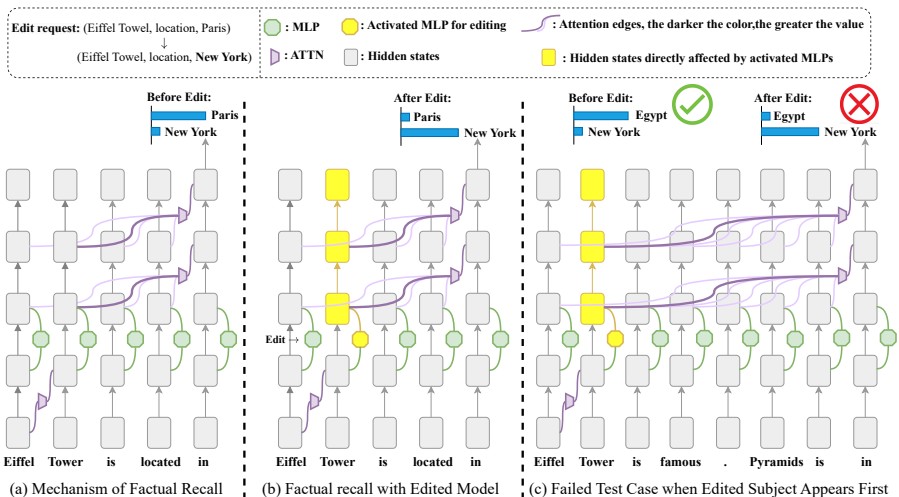

Figure 1: An illustration of counterfactual knowledge editing, where the new factual association (*Eiffel Tower, is located in, New York*) is edited in GPT-J-6b using the ROME method. (a) The hidden states of the subject are enriched by the MLP are successfully retrieved by the attention modules. (b) The editing method modifies the MLP parameters to alter the factual association. (c) The edited MLP generates hidden states that are prone to being mistakenly focused on by the attention modules.

*in "Egypt"*. Based on our preliminary study, we observe that a 6B GPT-J model after the knowledge editing can exhibit severe Specificity Failure in over 50% of cases regarding factual statements.

To delve deeper, the essence of the aforementioned issue fundamentally lies in the transmission of erroneous information flows that occur during the model's knowledge association recall process (Geva et al., 2023), where the model's prediction of certain knowledge can be viewed as the construction of a causal graph (Meng et al., 2022). Consequently, to ascertain the cause of Specificity Failure, we employ a lens experiment to scrutinize the causal graph within the model's generative process. Our observations reveal that the output attention activation at the last token position markedly contaminates the forward pass of the edited model, thereby resulting in erroneous outputs. Intuitively, the attention errors in the edited model could stem from the attention module's potential to erroneously focus on edited information, thus overlooking other pertinent details throughout the prediction process. By conducting the significance analysis and patching experiments on the attention module within the edited model, we observe an **Attention Drift phenomenon: the edited model assigns excessive attention scores to the entities related to the edited knowledge, thereby overly concentrating on the specific snippet within the context.** Consequently, this leads to outputs that predominantly align with the entities associated with the edited knowledge rather than conforming to the contextual semantics, even at the risk of clashing with pre-existing knowledge inherent in the model.

To alleviate the Attention Drift phenomenon, we propose a simple yet efficient strategy called *Selective Attention Drift Restriction* (**SADR**), which prevents excessive editing by constraining partial attention heads that overly focus on entities related to the edited knowledge. Specifically, we first locate the attention heads that exhibit severe Attention Drift phenomenon by comparing the model attention outputs before and after the editing, and then align the attention outputs of these identified heads to closely approximate those observed before editing. Currently, knowledge editing methods can be categorized into three types (Yao et al., 2023): locate-then-edit, parameter-preserving, and meta-learning approaches. We observe that severe specificity failure exists across methods from all three categories, with the accuracy of the original knowledge decreasing by more than half. Our proposed SADR can significantly mitigate specificity failure on five models ranging from 1.1B to 20B, and five editing methods covering all three categories, achieving improvements of up to 130.9% and 295.8% in the two main specificity tasks, with only a minimal 0.19% decrease in edit success.

## 2 NOTATION AND BACKGROUND

### 2.1 DEFINITION AND EVALUATION OF KNOWLEDGE EDITING

Previous works (Dai et al., 2021; Meng et al., 2022; 2023) represent factual associations as a knowledge tuple $t = (s, r, o)$, where $s$ is the subject, $r$ is the relation, and $o$ is the object. To evaluate

whether a factual association (e.g., *Eiffel Tower, is located in, Paris*) is captured in a LLM, we provide a prompt consisting of $s$ and $r$ (e.g., "Eiffel Tower is located in") and evaluate the model's prediction of $o$. Knowledge editing aims to replace the factual association stored in model parameters with a new factual association $(s, r, o_{edit})$, where $o_{edit}$ is the counterfactual target object (e.g., *New York*). In contrast, we denote $o_{true}$ as the true answer in the real world. Existing works (Meng et al., 2022; 2023; Mitchell et al., 2021; Tan et al., 2023) mainly evaluate knowledge editing in terms of *reliability*, *generalization* and *specifictiy* with the following metrics:

1) *Efficacy Score (ES)* and *Efficacy Magnitude (EM)* represent the portion of cases that $P(o_{edit}|s, r) > P(o_{true}|s, r)$ and the mean $P(o_{edit}|s, r)$, respectively.

2) *Paraphrase Score (PS)* and *Paraphrase Magnitude (PM)* evaluate generalization performance for a paraphrased prompt $(s_{para}, r_{para})$ (e.g., "The Tower of Eiffel stands in"). These two metrics can be written as $P(o_{edit}|s_{para}, r_{para}) > P(o_{true}|s_{para}, r_{para})$ and $P(o_{edit}|s_{para}, r_{para})$.

3) *Neighborhood Score (NS)* and *Neighborhood Magnitude (NM)* evaluate specificity by providing a neighboring but distinct subject $s'$ (e.g., "The Louvre"). These two metrics can be represented as $P(o_{true}|s', r) > P(o_{edit}|s', r)$ and $P(o_{true}|s', r)$, respectively.

As recent studies (Hoelscher-Obermaier et al., 2023; Yao et al., 2023) show that the presence of edited subjects in the inference context can deteriorate specificity performance, we incorporate two additional specificity metrics that include the edited subject in the test prompts.

1) *Relation Score (RS)* and *Relation Magnitude (RM)* evaluate how the model handles attributes of the edited subject that are unrelated to the edit, identified by the relation $r'$. Empirically, we find the edited model tends to predict the object $o_{edit}$ for even unrelated relations, e.g., "The color of Eiffel Tower is New York". Therefore, we calculate the Relation Score by $P(o_{true}|s, r') > P(o_{edit}|s, r')$ and the Relation Magnitude by $P(o_{true}|s, r')$.

2) *Distract Neighborhood Score (DNS)* and *Distract Neighborhood Magnitude (DNM))* function similarly to NS and NM but concatenate the edited sentence $(s, r, o_{edit})$ before the test prompt in the neighborhood task. These metrics can be written as $P(o_{true}|(s, r, o_{edit}) \oplus (s', r)) > P(o_{edit}|(s, r, o_{edit}) \oplus (s', r))$ and $P(o_{true}|(s, r, o_{edit}) \oplus (s', r))$, respectively.

We provide an example in Figure 2 to illustrate how to evaluate knowledge editing. Although knowledge editing involves various settings, such as batch (Meng et al., 2023) and sequential editing (Hartvigsen et al., 2024), we find that even editing a single factual association can significantly damage specificity performance when the edited subject occurs in the context. We call this **Specificity Failure** and focus on knowledge editing that modifies one factual association in this work.

### 2.2 THE KNOWLEDGE EDITING FRAMEWORK

To illustrate how knowledge editing methods work, we describe ROME (Meng et al., 2022) here as it is a foundational method that inspired subsequent techniques such as MEMIT (Meng et al., 2023), PMET (Li et al., 2024), and others (Li et al., 2024; Ma et al., 2023).

---

**CounterFact Example**

**Request Editing:** (Eiffel Tower, is located in, Paris) → (Eiffel Tower, is located in, New York)
**Editing Prompt:** Eiffel Tower is located in
**Editing Target:** New York

---

**Efficacy Task (ES):** Eiffel Tower is located in
**Paraphrase Task (PS):** Eiffel Tower stands in
**Neighborhood Task (NS):** The Louvre Museum is located in
**Relation Task (RS):** The color of Eiffel Tower is
**Distract Neighborhood Task (DNS):** Eiffel Tower is located in New York. The Louvre Museum is located in

Figure 2: Example of evaluation tasks for Knowledge Editing.

Geva et al. (2021) reveals that MLP layers can serve as two-layer key-value memories, with the output of the first MLP layer serving as $k$, and the output from the second layer acting as $v$, thereby facilitating knowledge retrieval about entities. Inspired by this, ROME changes $v$ to facilitate the model's prediction of $o_{edit}$ when $k$ is associated with the target subject. Meanwhile, it preserves $v$ as much as possible when $k$ is not related to the target subject, ensuring stability across unrelated contexts. To achieve this, ROME implements a rank-one update on weight $W$ of the second MLP layer with these two main objectives:

1 Minimize $\|\hat{W}k - Wk\|$ when $k$ is not from the last token of the target subject.

2 Satisfy $\hat{W}k^* = v^*$ when $k^*$ corresponds to the output of the last token of the target subject, while

$$v^* = \arg\min_z \left( -\log P_{G(m_t^{l^*} := z)} \left( o_{edit}|s,r \right) + \omega D_{\text{KL}} \left( P(x|p') \| P_{G(m_t^{l^*} := z)}(x|p') \right) \right).$$

Herein, the second objective seeks a vector $z$ that, when substituted as the output of the MLP in layer $l$ at token $t$ (denoted $G(m_t^{l^*} := z)$), it will lead the model to predict $o_{edit}$. $t$ is the last token of the subject. The KL divergence term minimizes the distances of predicted distribution between the predicted distributions for the prompt $p'$ (formatted as "subject is a") before and after editing, which controls the essence drift (Meng et al., 2022). $\omega$ is denoted as the controlling weight. ROME integrates these objectives by solving a linear system.

The general framework of locate-then-edit knowledge editing mentioned above can be viewed as first optimizing a certain vector $v^*$ that facilitates predicting the new knowledge, then integrating the vector $v^*$ into the model's parameters. We primarily focus on locate-then-edit knowledge editing in this paper, analyzing the causes of specificity failure within this framework. Additionally, we also evaluate specificity failure and the effectiveness of our SADR method on other parameter-preserving and meta-learning approaches.

## 3 EXPLORATION OF SPECIFICITY FAILURE

While knowledge editing excels at memorizing new knowledge, it still suffers from specificity failure. We first measure these failures on CounterFact benchmarks and then identify which intermediate outputs during the edited model's inference cause incorrect predictions. We further explore the primary triggers for these errors at a granular level and verify our findings by patching attention drift to mitigate specificity failure. Our experiments focus on *Relation* and *Distract Neighborhood* tasks, employing the widely-used ROME method on the GPT-J-6b model (Wang & Komatsuzaki, 2021).

### 3.1 CALIBRATING SPECIFICITY FAILURE ON COUNTERFACTUAL BENCHMARKS

To measure performance on both the *Relation* and *Distract Neighborhood* tasks, we employ a dataset composed of COUNTERFACT (Meng et al., 2022) and **WikiData**$_{counterfact}$ (Zhang et al., 2024a). The dataset includes 1683 factual statements, with more details provided in Appendix C.1.

Table 1: Editing results on GPT-J, whereas **red** numbers indicate a significant failure.

| Editor | Rewrite | | Generalization | | Specificity | | | | | |
|---|---|---|---|---|---|---|---|---|---|---|
| | ES ↑ | EM ↑ | PS ↑ | PM ↑ | NS ↑ | NM ↑ | RS ↑ | RM ↑ | DNS ↑ | DNM ↑ |
| None | 20.86 | 0.64 | 17.70 | 0.40 | 82.43 | 6.18 | 79.73 | 8.83 | 61.99 | 13.81 |
| ROME | 99.88 | 99.39 | 99.58 | 80.26 | 79.45 | 6.04 | **11.94** | **3.29** | **30.42** | **10.45** |
| MEMIT | 99.94 | 96.79 | 99.52 | 62.42 | 82.52 | 10.38 | **17.44** | **5.36** | **30.55** | 14.91 |
| PMET | 99.40 | 91.03 | 92.67 | 54.75 | 81.49 | 6.22 | **27.68** | **5.01** | **39.79** | 12.66 |

Table 1 illustrates a significant specificity failure when the edited subject occurs in the context, with the edited model incorrectly outputting the edited object in over 50% of test cases. Furthermore, the average probability of incorrect answers $o_{edit}$ is much greater than that of correct answers $o_{true}$, as 48.4% versus 3.3% for the *Relation* task and 24.9% versus 10.5% for the *Distract Neighborhood* task when editing with ROME. Editing methods such as MEMIT (Meng et al., 2023) and PMET (Li et al., 2024) also display significant specificity failure. This failure is further observed across a range of data formats and tasks, with detailed results provided in Appendix E.3.

### 3.2 LOCALIZING SPECIFICITY FAILURE

In the forward pass of an autoregressive language model, the flow of information can be viewed as a causal graph (Li et al., 2023b). When a model with $L$ layers predicts based on a prompt containing $T$ tokens, each module such as attention modules, MLPs, and transformer blocks, produces $T \times L$ outputs. Each of these outputs is influenced by prior outputs from earlier layers and preceding token positions. Inspired by causal tracing (Meng et al., 2022), we trace across different states in the causal graph to identify which parts contaminate the information flow in specificity failure.

We first conduct a forward pass on the edited model with test prompts and record outputs from various network modules across different layers and token positions. Then, we execute a forward pass with the vanilla model, copying the representations of specific modules from the stored outputs to the corresponding location without altering other computations. We traverse modules within a window of $k$ layers for each $l$-th layer and token position $t$. We refer to this approach as "Contaminating Substitution" and quantify its impact on the final output probability, formulated as:

$$\text{Tracing Effect} = P_{G(module_t^l := z)}(o_{true}|(s, r, o_{edit}) \oplus (s', r)) - P(o_{true}|(s, r, o_{edit}) \oplus (s', r)),$$

where $G(module_t^l := z)$ denotes the substitution of $z$ for the output of modules at token $t$ in layer $l$.

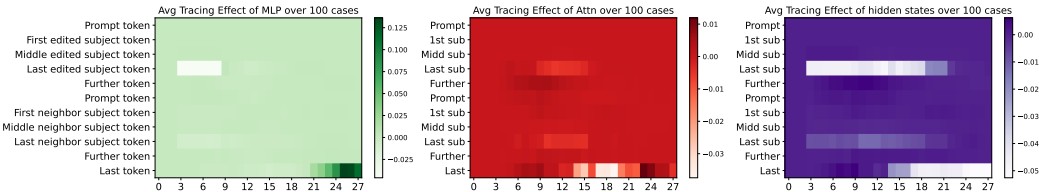

Figure 3: Visualizing "Contaminating Substitution" when replacing the output of specific modules on test prompts of the *Distract Neighborhood* task with window size 6.

As shown in Figure 3, the light-colored areas represent the primary states that lead to incorrect answers. We observe that replacing six layers of MLP activations or Attention activations can decrease the probability of a correct answer by up to 4.59% and 3.74%, respectively, while the total decrease caused by the edited model is 5.26%. As we edit the MLP module in the 5th layer, it is expected that the contaminating substitution of MLP activations near the edited layer significantly influences the final predictions. However, modifying the middle-upper layers of attention activations also has a similar impact on the correct output, suggesting that the primary cause of specificity failure is the attention module mishandling the information at the last token due to the edits. The findings are consistent on the *Relation* task, with additional experiments exploring the tracing effects with varying window sizes and prediction probability of $o_{edit}$ detailed in Appendix D.1.

## 3.3 IDENTIFYING ATTENTION DRIFT AS A TRIGGER FOR SPECIFICITY FAILURE

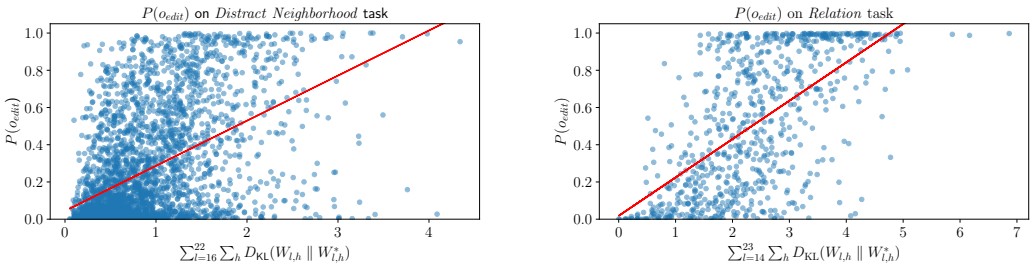

Figure 4: The correlation between the attention weight drift and $P(o_{edit})$ is positive on *Distract Neighborhood* ($\rho = 0.49; p < 1e-5$) and *Relation* ($\rho = 0.62; p < 1e-5$) tasks. The range of $l$ corresponds to the layers where attention activations have a significant impact on the final results.

As mentioned above, attention activations are one of the primary causes of specificity failures. Previous studies (Geva et al., 2023; Chen et al., 2024; Geva et al., 2022) have indicated that attention modules in the middle-upper layers extract factual attributes during predictions. This suggests that the attention module may mistakenly focus on edited information, thereby neglecting other information when predicting the final token. Therefore, we quantify the relationship between drift in attention weights and failures in the *Distract Neighborhood* and *Relation* tasks using the Pearson coefficient. Given that the edited model overestimates the probability of the edited object $o_{edit}$ relative to the true object $o_{true}$, we analyze the correlation between $\sum_l \sum_h D_{KL}(W_{l,h} \| W_{l,h}^*)$ and $P(o_{edit})$. Here,

$W_{l,h}$ and $W_{l,h}^*$ represent the attention weights that the last token attends to previous tokens in layer $l$ and head $h$ before and after editing, respectively.

Figure 4 shows a positive relationship between the drift in attention weights and the probability of the incorrect answer $o_{edit}$, suggesting that incorrect attention on previous information is a key factor for specificity failures. To further analyze the attention drift from the perspectives of attended tokens and heads, it's natural to figure out the following two questions:

1. The last token incorrectly allocates attention to previous tokens, leading to specificity failures. Among these previous tokens, which one has a significant impact when attended incorrectly?
2. Which has a greater impact on the prediction: the excessive localized attention drift of specific heads or the cumulative attention drift across all heads?

To address these questions, we calculate the Pearson coefficient between various factors and $P(o_{edit})$. Table 2 indicates that the Pearson coefficient for drift on the last token of the edited subject $s$ is higher compared to other tokens (Equation 1 vs Equation 2). Furthermore, the largest drift in attention weights among heads impacts the final result more than the cumulative drift across all heads (Equation 1 vs Equation 2), suggesting that the excessive attention of certain heads to the last subject token primarily triggers specificity failure.

Table 2: Linear correlation between factors related to attention weights and $P(o_{edit})$ with $p < 10^{-5}$.

| Tasks | Factors | $\rho$ |
|---|---|---|
| *Distract Neighborhood* | 1. $\sum_l \max_h W_{l,h}[s] - W_{l,h}^*[s]$ | 0.55 |
| | 2. $\sum_l \max_h \|W_{l,h}[\backslash s] - W_{l,h}^*[\backslash s]\|$ | 0.49 |
| | 3. $\sum_l \sum_h W_{l,h}[s] - W_{l,h}^*[s]$ | 0.43 |
| *Relation* | 1. $\sum_l \max_h W_{l,h}[s] - W_{l,h}^*[s]$ | 0.53 |
| | 2. $\sum_l \max_h \|W_{l,h}[\backslash s] - W_{l,h}^*[\backslash s]\|$ | 0.42 |
| | 3. $\sum_l \sum_h W_{l,h}[s] - W_{l,h}^*[s]$ | 0.50 |

## 3.4 MITIGATING SPECIFICITY FAILURE BY PATCHING ATTENTION DRIFT

To further verify the significant impact of attention drift on specificity failure, we quantify the change in prediction probability after patching attention weights across various layers. We first conduct a forward pass using the vanilla model with prompts from the specificity tasks and store the intermediate attention weights. Then, we test the edited model on the same prompts, substituting its attention weights with those previously stored.

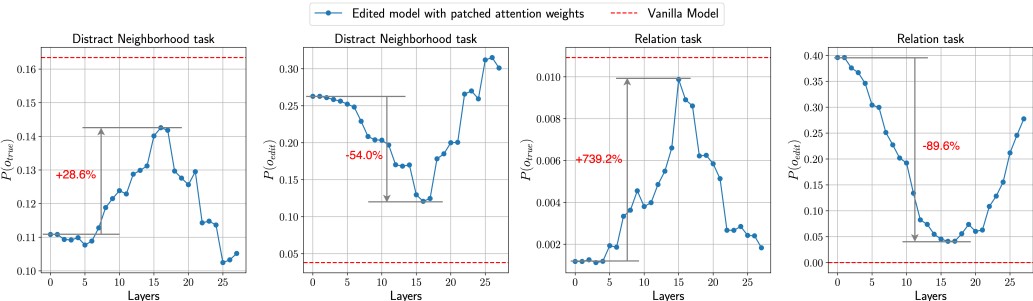

Figure 5: Impact of patching attention weights within a window size of 10 on the specificity tasks in the edited model. The performance of the original edited model is represented on the 0-th layer.

We find patching attention weights in middle-upper layers can lead to significant improvements in specificity tasks. As shown in Figure 5, patching attention weights in 10 consecutive layers result in a relative increase of 28.6% and 739.2% in the probability of the correct answer $P(o_{true})$, and a decrease of 54.0% and 89.6% in the probability of the wrong answer $P(o_{edit})$ for two specificity tasks, respectively. This shows preventing attention drift can effectively mitigate specificity failure.

## 3.5 TAKEAWAYS

Based on the analysis mentioned above, we can conclude that: **(1)** the attention activations at the last token position significantly contaminate the forward pass of the edited model, causing specificity

failures; **(2)** the max attention drift at the edited token position among heads is a primary trigger for the incorrect output $o_{edit}$, and **(3)** patching attention drift can largely mitigate the specificity failure.

# 4 SELECTIVELY RESTRAINING ATTENTION DRIFT DURING KNOWLEDGE EDITING

As mentioned in Section 2.2, the optimized value $v_*$ for knowledge editing can be obtained through gradient descent based on the following objective:

$$\mathcal{L}(z) = \frac{1}{N}\sum_{j=1}^{N}\underbrace{-\log P_{G(m_i^{(l^*)}:=z)}(o_{edit}\mid x_j+p)}_{\text{(a) Maximizing } o_{edit} \text{ probability}} + \underbrace{\omega D_{\text{KL}}\left(P(x\mid p')\|P_{G(m_{i'}^{(l^*)}:=z)}(x\mid p')\right)}_{\text{(b) Controlling essence drift}}.$$

(1)

However, this objective may cause attention drift that leads to Specificity Failure. To enhance specificity in knowledge editing when optimizing $v^*$, we introduce the **S**elective **A**ttention **D**rift **R**estriction (**SADR**), which is a regularization term based on Equation 1. It is worth noting that SADR dynamically applies constraints to different heads as needed since Transformer models contain various knowledge-specific attention heads (Wang et al., 2022; Geva et al., 2023) that capture different factual associations. Additionally, SADR is a simple yet efficient method and can be flexibly adapted across various editing methods.

More concretely, as excessive attention to the edited subject of certain heads is strongly correlated with Specificity Failures, we apply SADR on heads where the last token overly focuses on the edited subject. We determine which heads to restrain by the following criterion: **a head is selected if the attention weight attending to the subject's last token exceeds the maximum attention weight among all heads in the vanilla model**.

Let $W_{l,h}(S)$ be the attention weight from layer $l$ and head $h$ when processing the prompt $S$, $W_{l,h}^{G(m_i^{(l^*)}:=z)}(S)$ be the attention weight from the model that is edited with $z$, and $M_l(S) = \max_h W_{l,h}(S)[-1,s]$ be the maximum attention weight that the last token attends to the edited subject $s$ among all heads at layer $l$ in the vanilla model. The objective of SADR can be written as:

$$\mathcal{L}_{SADR}(z) = \frac{1}{N}\sum_{j=1}^{N}\sum_{l}\sum_{h\in H_l(S_j)}D_{\text{KL}}\left(W_{l,h}(S_j)[-1,:]\|W_{l,h}^{G(m_i^{(l^*)}:=z)}(S_j)[-1,:]\right), \quad (2)$$

where $H_l(S_j) = \{h : W_{l,h}^{G(m_i^{(l^*)}:=z)}(S_j)[-1,s] > M_l(S_j)\}$ and $S_j = x_j \oplus (s,r)$.

Thus, the optimized value $v_*$ can be obtained by: $v_* = \arg\min\left(\mathcal{L}(z) + \gamma\mathcal{L}_{SADR}(z)\right)$, where $\gamma$ is the controlling weight.

# 5 EXPERIMENTS

## 5.1 SETTINGS

**Dataset** Due to the limited availability of datasets that satisfy the required fields for our tasks, we combine COUNTERFACT (Meng et al., 2022) and WikiData$_{counterfact}$ (Zhang et al., 2024a) with 1,683 factual statements as the testing data. The processing details are mentioned in Appendix C.1. Additionally, we extend our experiments to broader datasets, including QA-format and recent knowledge editing tasks, as detailed in Appendix E.3. The phenomena of Specificity Failure and the performance of SADR remain consistent across these datasets.

**Baselines & Models** We evaluate the performance of our methods on three mainstream locate-then-edit knowledge editing baselines: ROME (Meng et al., 2022), MEMIT (Meng et al., 2023), and PMET (Li et al., 2024). Specifically, we focus on knowledge editing with one factual association for all the baselines. We implement our SADR method across three editing baselines on the GPT-J-6b (Wang & Komatsuzaki, 2021), Llama3-8b (AI@Meta, 2024) and GPT-NeoX-20b (Black et al.,

2022) models. In Appendix E.1 and E.2, we also conduct SADR on more model variants and editing methods, including parameter-preserving and meta-learning approaches. More details about baseline methods and our implementations can be found in Appendix C.2.

**Metrics** Apart from the metrics mentioned in Section 2.1, we utilize Fluency Score (FL) (Meng et al., 2022) to evaluate the generation ability of the edited model with prompts related to the edited subject, which is computed by the weighted mean of *bi*-gram and *tri*-gram entropies. Results closer to that of the vanilla model indicate better performance. To further test the generalization–specificity tradeoff, we report the harmonic mean of ES, PS, NS, RS, and DNS as the average score (Avg. S). We also test the model's commonsense reasoning abilities and its perplexity in language modeling, with results reported in the Appendix E.1.

## 5.2 MAIN RESULTS

Table 3: Results of our methods across three edit methods. The values for the 95% confidence intervals are displayed in parentheses. **Bold** number indicates better performance and **Green** number indicates a significantly better score with more than 50% relative improvement.

| Model | Editor | Score | Rewrite | Generalization | Specificity | | | Fluency |
|---|---|---|---|---|---|---|---|---|
| | | Avg. S ↑ | ES ↑ | PS ↑ | NS ↑ | RS ↑ | DNS ↑ | FL |
| GPT-J (6b) | None | 34.43 | 20.86(2.0) | 17.70(1.8) | 82.43(1.0) | 79.73(2.1) | 61.99(1.3) | 621.96(0.9) |
| | ROME | 33.56 | **99.88(0.2)** | **99.58(0.3)** | 80.26(1.0) | 11.94(1.7) | 30.42(1.2) | 620.58(1.2) |
| | + ours | **57.74** | 99.76(0.2) | 96.36(0.9) | **80.86(1.0)** | **27.75(2.3)** | **49.35(1.3)** | **623.00(1.0)** |
| | MEMIT | 51.07 | **99.64(0.3)** | **95.47(1.0)** | 81.42(1.0) | 25.78(2.3) | 38.00(1.2) | 622.71(1.0) |
| | +ours | **60.23** | 99.52(0.3) | 93.21(1.2) | **81.44(1.0)** | **34.68(2.5)** | **47.35(1.3)** | **623.60(0.9)** |
| | PMET | 52.98 | **99.40(0.4)** | **92.67(1.3)** | **81.49(1.0)** | 27.68(2.3) | 39.79(1.3) | 621.18(1.1) |
| | +ours | **59.06** | 99.11(0.5) | 89.09(1.5) | 81.44(1.0) | **33.33(2.5)** | **47.47(1.3)** | **622.18(1.0)** |
| Llama3 (8b) | None | 19.99 | 9.36(1.4) | 9.48(1.4) | 87.17(0.9) | 92.66(1.4) | 64.25(1.2) | 617.19(1.3) |
| | ROME | 58.54 | **99.88(0.2)** | **99.52(0.3)** | 82.19(1.0) | 29.38(2.4) | 52.21(1.3) | **617.23(1.8)** |
| | + ours | **70.99** | 99.82(0.2) | 96.90(0.8) | **83.18(1.0)** | **47.67(2.6)** | **58.51(1.3)** | 618.39(1.7) |
| | MEMIT | 40.90 | **99.94(0.1)** | **99.52(0.3)** | 82.52(1.0) | 17.44(2.0) | 30.55(1.2) | 605.99(4.3) |
| | +ours | **63.12** | 99.82(0.2) | 98.87(0.5) | **82.96(1.0)** | **35.38(2.5)** | **45.57(1.3)** | **618.38(1.9)** |
| | PMET | 43.87 | **99.58(0.3)** | **99.40(0.4)** | 81.10(1.0) | 19.84(2.1) | 32.12(1.2) | 610.65(3.4) |
| | +ours | **61.88** | 99.28(0.4) | 97.02(0.8) | **82.86(1.0)** | **34.68(2.5)** | **51.22(1.3)** | **617.91(1.6)** |
| GPT-NeoX (20b) | None | 31.87 | 17.04(1.8) | 17.64(1.8) | 80.62(1.0) | 83.76(1.9) | 58.34(1.3) | 619.25(0.9) |
| | ROME | 23.45 | **99.94(0.1)** | **98.75(0.5)** | 72.67(1.1) | 15.11(1.9) | 8.84(0.7) | 579.82(6.6) |
| | + ours | **54.81** | 99.76(0.2) | 96.13(0.9) | **73.96(1.1)** | **34.89(2.5)** | **34.95(1.2)** | **619.54(1.1)** |
| | MEMIT | 40.32 | **99.88(0.2)** | **90.46(1.4)** | 77.45(1.1) | 35.24(2.5) | 16.22(0.9) | 615.19(2.2) |
| | +ours | **45.26** | 97.38(0.8) | 89.21(1.5) | **77.45(1.1)** | **45.48(2.6)** | **18.49(1.0)** | **621.59(0.8)** |
| | PMET | 19.97 | **99.52(0.3)** | **95.23(1.0)** | 74.18(1.1) | 12.29(1.7) | 7.41(0.7) | 510.25(9.8) |
| | +ours | **31.16** | 99.40(0.4) | 93.09(1.2) | **75.33(1.1)** | **24.86(2.3)** | **11.61(0.8)** | **589.76(5.5)** |

**Specificity Failure is prevalent in existing knowledge editing methods.** As shown in Table 4, the *Relation* and *Distract Neighborhood* tasks exhibit a significant decline across all editing methods, even though these methods perform edits at different layers and modules. Specifically, in the *Relation* task, the accuracy of all edited models drops to less than half of their original performance. This indicates that specificity failure is a widespread and severe issue in knowledge editing.

**SADR significantly mitigates Specificity Failure.** In the *Relation* and *Distract Neighborhood* tasks, SADR consistently improves the specificity across all editing methods. Notably, in over half of the setups, our method enhances the original specificity metrics by more than 50% (marked in green). SADR also stabilizes the performance in the *Neighborhood* task and the *Fluency* of generated text, e.g., when using SADR on ROME and PMET with GPT-NeoX, the *Fluency* score is significantly improved. SADR can also achieve better performance on TinyLlama-1.1b (Zhang et al., 2024b) and Llama2-13b (Touvron et al., 2023) compared with baselines, indicating that our method is effective across models with various sizes and knowledge densities. We provide more detailed results in Appendix E.1 and report the human evaluation results in Appendix E.4.

**SADR has minimal impact on edit success rates.** SADR results in less than a 3% decrease in performance on the *Rewrite* and *Generation* tasks. It is important to note that significantly mitigating specificity failure while fully preserving rewrite and generalization performance is quiet difficult. This is because in many previous model editing evaluation frameworks, the specificity failure highlighted in our paper is often overlooked. As a result, prior methods tend to prioritize generalization and rewrite scores, neglecting the risks of specificity failure. Under such evaluation criteria, high scores can be achieved by simply identifying the subject and greedily predicting the object, even if the relationship between them is completely ignored.

We believe that ensuring stable knowledge editing is more important than achieving nearly 100% accuracy in generalization. In real-world scenarios, methods that achieve 97% generalization with stable and safe edits are often more acceptable than those with 100% generalization but significant specificity failures (e.g., severe knowledge errors caused by attention drift after editing the subject). Furthermore, while addressing specificity failure without compromising generalization is challenging, we demonstrate a better balance between these two aspects in Section 6.2.

# 6 ABLATION STUDY

## 6.1 EFFECT OF RESTRAINING HEADS SELECTION

We first explore the effects of selectively restraining heads that exhibit significant attention drift compared to restraining all heads across various control weights $\gamma$ on ROME with GPT-J. Edit Success is quantified by the average of *PS* and *ES*, while Specificity is calculated as the average of *NS*, *RS*, and *DNS*. As shown in Figure 6, selectively restraining heads that overfocus on the edited token outperforms restraining all heads on both the Edit Success and Specificity across different $\gamma$ settings. This suggests that not all drift in attention is harmful; rather, it is excessive attention compared to the vanilla model that should be addressed.

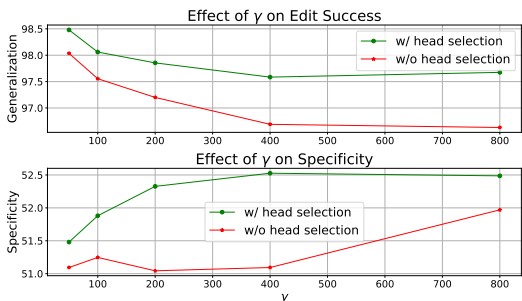

Figure 6: Impact of selective head restriction on Edit Success and Specificity performance

## 6.2 TRADE-OFF BETWEEN GENERALIZATION AND SPECIFICITY

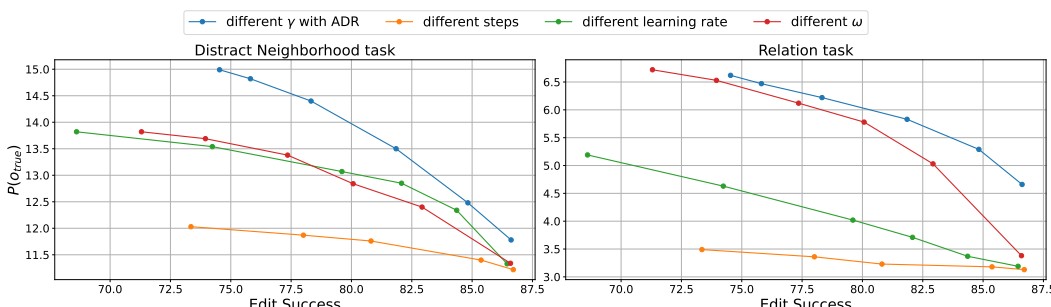

Figure 7: Analysis of trade-offs by adjusting different hyperparameters.

Knowledge editing methods present a trade-off between Edit Success and Specificity, which can be visualized through adjustments in hyperparameters such as optimization steps, learning rate, and $\omega$ which controlling the essence drift. We analyze the trade-off of our method by varying $\gamma$ and compare it with the trade-offs from adjusting other hyperparameters in the original ROME on GPT-J. As changes are more clear on $P(o_{edit})$ than the proportion of cases which $P(o_{edit}) > P(o_{true})$, we use the average of *EM* and *PM* to measure Edit Success and apply *RM* and *DNM* for evaluating *Relation* and *Distract Neighborhood* task, respectively. Further details are illustrated in Appendix C.3.

Figure 7 shows that our method exhibits a superior trade-off compared to the adjustments of hyperparameters in the original ROME method, indicating that **SADR** can obtain $v^*$ that enable the model to more effectively distinguish when to output the edited knowledge.

## 7 RELATED WORK

**Knowledge editing.** The field of knowledge editing has recently emerged, aiming to modify model knowledge at a low cost without adversely affecting performance. The methods can be categorized into three main paradigms: parameter-preserving, locate-then-edit, and meta-learning approaches (Yao et al., 2023; Wang et al., 2023; Mazzia et al., 2023). Parameter-preserving methods explicitly store modified knowledge in memory and use techniques such as classifiers (Mitchell et al., 2022), prompt engineering (Madaan et al., 2022; Zhong et al., 2023; Zheng et al., 2023), or external parameters (Dong et al., 2022; Huang et al., 2023; Hartvigsen et al., 2024; Wang et al., 2024b) to retrieve the knowledge. locate-then-edit methods update specific parameters by identifying where the targeted knowledge is stored and directly editing those locations (Meng et al., 2022; 2023; Li et al., 2024; Ma et al., 2023). Meta-learning approaches involve training a hypernetwork to edit the model's knowledge parameters (Mitchell et al., 2021; Tan et al., 2023).

Despite the promising prospects of knowledge editing, significant challenges remain in specificity and generalization. Editing specific pieces of knowledge can lead to ripple effects within the knowledge graph (Wang et al., 2024a; Cohen et al., 2024), but the edited model may be prone to under-editing (Li et al., 2023d; Pinter & Elhadad, 2023) or over-editing (Li et al., 2023c) in response to these changes. Recent works (Gu et al., 2024; Hazra et al., 2024; Yang et al., 2024b) also find that knowledge editing may impair general abilities as the number of edits increases. While an ideal edited model should generalize the impacts of edits within a knowledge graph and safely conduct large-scale modifications, Hoelscher-Obermaier et al. (2023) and Rosati et al. (2024) identify a more pressing issue in knowledge editing: the performance of the model drops dramatically when the edited subject or sentence appears in the context. This motivates us to focus on this issue in the current paper.

**Mechanisms of Factual Associations in Transformers.** Transformer (Vaswani et al., 2017) is the most commonly used architecture for large language models, achieving remarkable performance attributed to the vast amount of knowledge stored in its parameters (Petroni et al., 2019; Roberts et al., 2020; Cao et al., 2023). Significant efforts have been made to uncover the mechanism of how transformers memorize and retrieve knowledge during training and inference. Geva et al. (2021) view the FFN layers as the primary module that stores knowledge in a key-value format. Dai et al. (2021) and Geva et al. (2022) explore the manipulation of factual associations by intensifying or attenuating the values in the activated outputs of the first FFN layer. Meng et al. (2022) employ causal tracing to demonstrate the crucial role that early MLP layers play in factual recall. Hase et al. (2023) find that editing on layers where knowledge is not primarily stored can also achieve a high success rate, indicating that it is possible to "override" the information in layer $l$ with an edit to another layer $k$. Additionally, Hao et al. (2021) find that the self-attention module performs attribute extraction during factual association recall. Building on the knowledge recall mechanisms identified in previous works, we explore the reasons behind Specificity Failure and design our approach to mitigate attention drift.

## 8 CONCLUSION

In this paper, we explore and mitigate the limitations of current knowledge editing methods in Large Language Models (LLMs) when the edited content is present in the context. Our investigations reveal the **Specificity Failure** issue that the attention mechanisms in these models overly focus on the edited entities, leading to a neglect of other relevant information in the context. Based on the preliminary study, such a Specificity Failure issue stems from the **Attention Drift** phenomenon: the edited model assigns excessive attention scores to the entities related to the edited knowledge, thereby overly concentrating on specific snippets within the context. Thus, based on the previous knowledge editing methods, we propose the **S**elective **A**ttention **D**rift **R**estriction (**SADR**) method, which introduces a regularization term during the editing process, dynamically constraining the weight of partial selected attention heads and preventing excessive focus on the edited entities. Our experiment indicates that SADR can significantly reduce Specificity Failures while preserving high rates of editing success.

## REPRODUCIBILITY STATEMENT

Our work is based on open-source models and datasets. In Section 5 and Appendix C, we provide detailed descriptions of data processing, method implementation, and ablation experiments. Additionally, in the supplementary materials, we include the complete code for our method as well as the processed datasets.

## ACKNOWLEDGMENTS

We want to thank all the anonymous reviewers for their valuable comments. This work was supported by the National Science Foundation of China (NSFC No. 62206194), the Natural Science Foundation of Jiangsu Province, China (Grant No. BK20220488), the Young Elite Scientists Sponsorship Program by CAST (2023QNRC001), and the Priority Academic Program Development of Jiangsu Higher Education Institutions.

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

# A    LIMITATIONS AND FUTURE WORKS

In this section, we outline several limitations of our study that highlight areas for future research and improvement: **(1)** While our method shows promise, there is still potential for improvement in the Relation task. This may be due to the fact that edits can inadvertently erase or obscure other relevant knowledge about the subject, thereby affecting the model's overall performance in understanding and maintaining relations. **(2)** Our study primarily focused on single factual association edits. This scope excluded more complex scenarios such as batch and sequential editing, which involve multiple edits either simultaneously or over a sequence. **(3)** We focus on knowledge editing on only transformer-based models, omitting models with new architectures (Gu & Dao, 2023). These issues highlight important avenues for future research and will be explored in subsequent studies to enhance the robustness and applicability of knowledge editing.

# B    PRELIMINARY OF KNOWLEDGE EDITING

In this section, we provide more details of the baselines used in our experiments.

**ROME**    As mentioned in Section 2.2, ROME (Meng et al., 2022) implements rank-one knowledge editing by deriving a closed form solution:

$$\text{minimize } \|\hat{W}K - V\| \text{ such that } \hat{W}k_* = v_* \quad \text{by setting } \hat{W} = W + (v_* - Wk_*)\frac{(C^{-1}k_*)^T}{(C^{-1}k_*)^T k_*},$$

$$(3)$$

where $C = KK^T$ is a constant that estimates the uncentered covariance of $k$ from samples of Wikipedia text. In order to choose the lookup key $k^*$ for the subject, ROME concatenates different prefixes generated by the vanilla model with the sentence of the editing request. Then, ROME records the activations of the first layer of MLP, consider the average value $k_* = \frac{1}{N}\sum_{j=1}^{N} k(x_j + s)$ as the lookup key. The $v^*$ for recalling the fact is obtained by minimizing the objective in Equation 1.

**MEMIT**    In order to directly update multiple memories in a language model, MEMIT employs batch update and multiple layers update based on ROME. To derive an optimal single-layer update that minimizes the squared error of memorized associations while preserving existing memories, the expanded objective for batch update is defined as:

$$W_1 \triangleq \underset{\hat{W}}{\arg\min} \left( \sum_{i=1}^{n} \left\| \hat{W}k_i - m_i \right\|^2 + \sum_{i=n+1}^{n+u} \left\| \hat{W}k_i - m_i \right\|^2 \right),$$

$$(4)$$

where $W_1$ is the new matrix for the second layer of the FFN. Here, $K_0 = [k_1 \mid k_2 \mid \cdots \mid k_n]$ and $M_0 = [m_1 \mid m_2 \mid \cdots \mid m_n]$ represent the original keys and memories in the vanilla $W_{out}^l$, while $K_1 = [k_{n+1} \mid k_{n+2} \mid \cdots \mid k_{n+u}]$ and $M_1 = [m_{n+1} \mid m_{n+2} \mid \cdots \mid m_{n+u}]$ are new $u$ factual associations that need to be edited in the model.

By solving the linear system, the updated matrix $W_1$ can be formalized as:

$$W_1 = W_{out}^l + RK_1^T(C_0 + K_1K_1^T)^{-1},$$

$$(5)$$

where $C_0$ is the aggregate statistic over the previously stored keys, computed by a random sample of inputs as in ROME.

In order to improve the robustness, MEMIT distributes updates evenly across the range of mediating layers $\mathcal{R}$. The approach to obtain $k^*$ and $v^*$ is similar to ROME. MEMIT calculates $\delta^l$ in ascending layer order to prevent the influence of edit layers on subsequent layers. The update process of MEMIT can be represented by Algorithm 1.

---

**Algorithm 1:** The MEMIT Algorithm

---

**Data:** Requested edits $\mathcal{E} = \{(s_i, r_i, o_i)\}$, generator $G$, layers to edit $\mathcal{R}$, stored keys $C_0^l$
**Result:** Modified generator containing edits from $\mathcal{E}$

1 **for** $s_i, r_i, o_i \in \mathcal{E}$ **do**             // Compute target vectors $v_i$ for each memory $i$

2     **optimize** $\delta_i \leftarrow \sum_{j=1}^{N} - \log P_{G(h_i^{(l*)} += \delta_i)}(o^* \mid x_j + p)$

3     $v_i \leftarrow h_i^{(L)} + \delta_i$

4 **end**

5 **for** $l \in \mathcal{R}$ **do**             // Perform update over layers in ascending order

6     $h_i^{(l)} \leftarrow transformer\_block(h_i^{l-1})$    // Execute layer $l$ using the updated weights

7     **for** $s_i, r_i, o_i \in \mathcal{E}$ **do**

8         $k_i^l \leftarrow k_i^{(l)} = \frac{1}{N} \sum_{j=1}^{N} k(x_j + s_j)$

9         $r_i^l \leftarrow \frac{v_i - h_i^{(L)}}{L - l + 1}$

10     **end**

11     $K^l \leftarrow [k_i^{l_1}, ..., k_i^L]$

12     $R^l \leftarrow [r_i^{l_1}, ..., r_i^L]$

13     $\Delta^l \leftarrow R^l K^{l^T}(C_0^l + K^l K^{l^T})^{-1}$

14     $W^l \leftarrow W^l + \Delta^l$         // Update MLP weights in layer $l$

15 **end**

---

**PMET**   PMET (Li et al., 2024) discovers that Multi-Head Self-Attention(MHSA) weights do not require updating when new knowledge is introduced, thus only integrating the optimized FFN activation to conduct precise editing. PMET introduces optimizable parameters, $\delta_i^a$ for the MHSA output and $\delta_i^m$ for the FFN output at the $L$-th layer. It then retains only the optimized hidden states of the FFN to update its weights, represented as $v_i^m = m_i^L + \delta_i^m = \arg\min \mathcal{L}(z)$, where $\mathcal{L}(z)$ refers to the objective in Equation 1. Following this, PMET employs the same algorithmic steps as MEMIT to update the FFN weights.

## C   Implementation Details

### C.1   Dataset Processing

The dataset we use is a mixture of counterfact datasets from Meng et al. (2022) and Zhang et al. (2024a). Meng et al. (2022) introduce COUNTERFACT, which contains 21,919 records featuring a diverse set of subjects, relations, and linguistic variations. It also provides paraphrase prompts, neighborhood prompts, and generation prompts for specificity evaluation. Zhang et al. (2024a) collect triplets about popular entities from top-viewed pages on Wikipedia to construct **WikiData**$_{counterfact}$. They provide relational prompts to evaluate the impact of edits on other attributes associated with the edited subject. We combined these datasets in equal proportions to create a balanced dataset with 1683 factual statements.

When measuring specificity, it is crucial that the neighborhood subject and the test relationship of the subject remain unaffected by the edit. For example, when editing the factual knowledge tuple "(*Carl Bosch, citizenship, Germany*)" to "(*Carl Bosch, citizenship, Canada*)", it might be logically consistent to also generalize Carl Bosch's birthplace to Canada, which should not be considered in specificity tests. However, this edit should not alter the answer to "The gender of Carl Bosch is". Test cases like this should be considered in specificity tests. To measure specificity more accurately, we filter our dataset using GPT-4. The filtering prompts are detailed in Figs. 8.

### C.2   Baseline Settings

In this section, we detail the parameters used for each baseline and the SADR method across different models. We utilize ROME (Meng et al., 2022), MEMIT (Meng et al., 2023), and PMET (Li et al., 2024) as the baseline knowledge editing techniques, implemented using EasyEdit [2]. As the objective of SADR prevents over-editing by restricting optimization from causing large attention drift, it is

---

[2]https://github.com/zjunlp/EasyEdit

---

**Prompts for Filtering Specificity Cases**

**Request Editing:** Irma Boom spoke the language → Russian
**Test prompt for neighborhood:** Johannes Lingelbach is a native speaker of
**Test prompt for relationship:** The place of birth of Irma Boom is

---

**Prompt for gpt-4 to filter neighborhood:**
Determine the subjects in the following two sentences is related or unrelated.
- Neighborhood subjects in the same field without
direct collaboration or interaction are considered unrelated.
Sentence 1: {**Request Editing**}
Sentence 2: {**Test prompt for neighborhood**}
Expected answer: Related or Unrelated.

**Prompt for gpt-4 to filter relationship:**
Determine the factual relationships in the following two sentences is related or unrelated:
Sentence 1: {**Request Editing**}

Sentence 2: {**Test prompt for relationship**}
Expected answer: Related or Unrelated.

Figure 8: Prompts for gpt-4 to filter specificity cases

necessary to increase the number of optimization steps to ensure convergence. We test $[20, 40, 80]$ optimization steps with restraining weights $\gamma$ set at $[5e-3, 1e-2, 4e-2, 8e-2]$ on the validation split. All experiments are conducted on eight NVIDIA A100 (40GB) GPUs, with individual edits taking approximately 20 to 80 seconds on a single GPU. Completing all edits and evaluations across our dataset requires 1-2 days.

**ROME** The parameters applied for the original baseline are consistent with the original paper. The learning rate is 0.5, optimization steps are 20, and the KL factor $\omega$ is 0.0625 across various models. For GPT-J-6b, we edit layer 5, with optimization steps of 80 and a controlling weight $\gamma = 1e-2$ for the SADR method. For Llama3-8B, we edit layer 5, with optimization steps of 80 and a controlling weight $\gamma = 4e-2$ for the SADR method. For GPT-NeoX-20b, we edit layer 5, with optimization steps of 80 and a controlling weight $\gamma = 1e-2$ for the SADR method. For Llama2-13B, we edit layer 15, with optimization steps of 80 and a controlling weight $\gamma = 1e-2$ for the SADR method. For TinyLlama, we edit layer 4, with optimization steps of 80 and a controlling weight $\gamma = 8e-2$ for the SADR method.

**MEMIT** The original baseline applies learning rate of 0.5, 20 optimization steps, and KL factor $\omega$ of 0.0625 across various models. For GPT-J-6b, we edit layers 3-8, with optimization steps of 20, learning rate 0.25, and a controlling weight $\gamma = 5e-3$ for the SADR method. For Llama3-8B, we edit layers 4-8, with optimization steps of 80 and a controlling weight $\gamma = 1e-2$ for the SADR method. For GPT-NeoX-20b, we edit layers 13-16, with optimization steps of 80 and a controlling weight $\gamma = 5e-3$ for the SADR method. For Llama2-13B, we edit layers 5-9, with optimization steps of 80 and a controlling weight $\gamma = 4e-2$ for the SADR method. For TinyLlama, we edit layers 3-5, with optimization steps of 80 and a controlling weight $\gamma = 8e-2$ for the SADR method.

**PMET** The original baseline applies learning rate of 0.5, 20 optimization steps, and KL factor $\omega$ of 0.0625 across various models. For GPT-J-6b, we edit layers 3-8, with optimization steps of 40 and a controlling weight $\gamma = 8e-2$ for the SADR method. For Llama3-8B, we edit layers 4-8, with optimization steps of 80 and a controlling weight $\gamma = 1e-2$ for the SADR method. For GPT-NeoX-20b, we edit layers 13-16, with optimization steps of 20 and a controlling weight $\gamma = 2e-3$ for the SADR method. For Llama2-13B, we edit layers 7-9, with optimization steps of 20 and a controlling weight $\gamma = 5e-3$ for the SADR method. For TinyLlama, we edit layers 3-5, with optimization steps of 40 and a controlling weight $\gamma = 8e-2$ for the SADR method.

### C.3 ABLATION SETTINGS

We further illustrate the implementation details for experiments in Section 6. When testing the trade-off between generalization and specificity, we randomly sample 500 data points for evaluation. The controlling weight $\gamma$ for the SADR method is applied at $[1e-5, 1e-4, 5e-4, 2.5e-4, 1e-2, 8e-2]$.

The optimization steps are applied at $[7, 14, 20, 40, 80, 120]$. The KL factor $\omega$ is applied at $[0.1, 1, 2, 3, 4, 5]$. The learning rate is applied at $[5e-2, 6e-2, 8e-2, 1e-1, 2e-1, 4e-1]$.

# D ADDITIONAL RESULTS FOR EXPLORING SPECIFICITY FAILURES

## D.1 LOCALIZE SPECIFICITY FAILURE IN CAUSAL GRAPH

We further investigate the tracing effects of "Contaminating Substitution" across different window sizes in the "Distract Neighborhood" and "Relation" tasks, and also demonstrate the impact on the prediction probability of $o_{edit}$. As shown in Figure 9a, varying window sizes indicate similar areas



(a) Tracing Effect on $P(o_{true})$ with different window sizes in the *Distract Neighborhood* task.

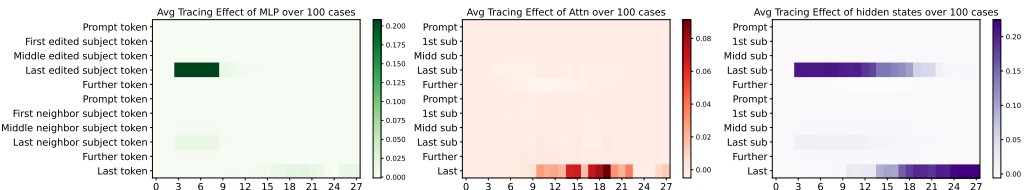

(b) Tracing Effect on $P(o_{edit})$ in the *Distract Neighborhood* task with window size 6.

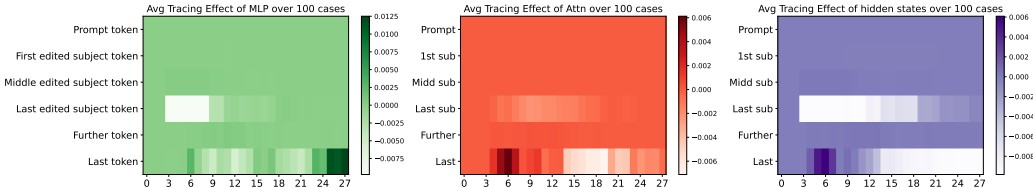

(c) Tracing Effect on $P(o_{true})$ in the *Relation* task with window size 6.

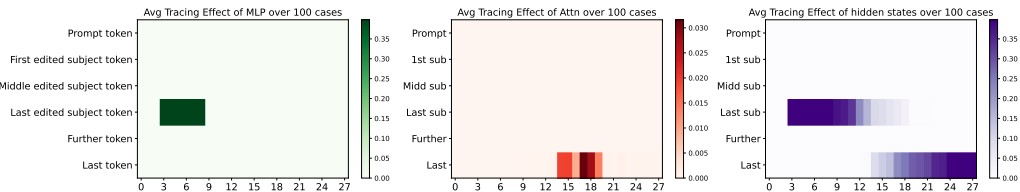

(d) Tracing Effect on $P(o_{edit})$ in the *Relation* task with window size 6.

Figure 9: Visualizing "Contaminating Substitution" with different window sizes, specificity tasks and prediction objects.

leading to Specificity Failures, and the decrease in $P(o_{true})$ is correlated with window size. This suggests that contaminating information accumulates at the last token in middle-upper layers due to the recall mechanism of attention modules.

By comparing Figure 3 with Figure 9b, and Figure 9c with Figure 9d, we observe notable similarities in the areas that correspond to increases in incorrect answer probabilities $P(o_{edit})$ and decreases in correct answer probabilities $P(o_{true})$. This suggests that the same information flow may be driving changes in both probabilities.

Furthermore, a phenomenon that may seem counterintuitive is that replacing MLP or Attn activations in the final layers increases the probability of correct answers. This can be attributed to the disruption of anti-overconfidence mechanisms in the final layers (Lv et al., 2024).

## D.2 PATCHING ATTENTION DRIFT TO MITIGATE SPECIFICITY FAILURE

In section 3.4, we demonstrate an improvement in specificity performance after patching attention drift in some consecutive layers. To explore the effectiveness of patching attention drift at a finer granularity, we evaluate the tracing effect of modifying a single value in the attention weight matrix. Specifically, we alter the value that represents the weight of the last token attending to the $t$-th token before softmax during the forward pass of the edited model. The replacement value is the one generated by the vanilla model for the same prompts. Considering residual connections in Transformer, we patch for a window of $k$ layers around the $l$-th layer.

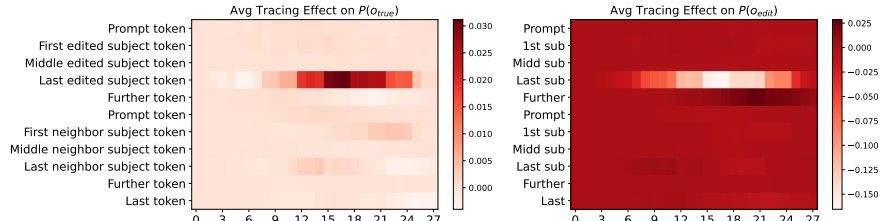

(a) Tracing Effect with window size 10 in the *Distract Neighborhood* task.

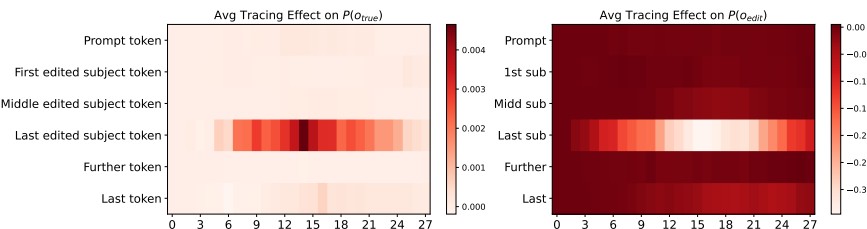

(b) Tracing Effect with window size 10 in the *Relation* task.

Figure 10: The tracing effect of patching the attention value that the last token attends to the previous tokens for different layers.

As shown in Figure 10, patching the attention value of the last edited subject token in the middle-upper layers significantly mitigates the Specificity Failure, where the magnitude of change in probability closely matches the one of replacing the entire attention weight matrix in Figure 5. This aligns with the findings in Section 3.3 that the drift of attention weights from the last token to the edited token is the main trigger for Specificity Failure.

## D.3 THE CORRELATION BETWEEN MORE FACTORS AND SPECIFICITY FAILURE

Recent works (Fang et al., 2024; Yao et al., 2024; Ma et al., 2024) point out that the edit vector's direction, space, and norm can influence the model's specificity performance. However, these works primarily focus on preserving general knowledge and capabilities, rather than addressing the specificity failure that arises when the edited subject appears in the context. To explore the relevance of these factors to the specificity failure problem studied in our work, we conducted a correlation analysis. Specifically, we compared four factors—attention drift, hidden

Table 4: Correlation of different factors with specificity failure.

| Factor | Pearson Coefficient (Distracting Neighborhood) | Pearson Coefficient (Relation) |
|---|---|---|
| **Attention Drift** | **0.49** | **0.62** |
| **Hidden State Norm** | 0.01 | 0.31 |
| **L2 Distance** | 0.01 | 0.31 |
| **Cosine Similarity** | 0.02 | -0.15 |

state norm post-editing, L2 distance between hidden states pre- and post-editing, and the cosine similarity of hidden states pre- and post-editing —with the probability of $P(o_{\text{edit}})$ in specificity tasks.

Table 4 shows that, compared to the direction or norm of the edit vector, attention drift has a more direct and significant impact on specificity failure.

### D.4    DISCUSSION ABOUT REASONS FOR ATTENTION DRIFT

Experiments have shown that attention drift is closely related to specificity failure. A natural question arises: what is the reason for attention drift during the editing process? Intuitively, editing methods primarily modify the hidden states of the edited subject, which subsequently influence the final output through the attention mechanism. In traditional editing methods (e.g., ROME discussed in Section 2.2), the optimization objective explicitly trains the model to predict the new $o_{\text{edit}}$ given $(s, r)$. This may create a shortcut, where the hidden state of the subject is shaped in a way that makes it overly prone to being prioritized by the attention mechanism, thereby hard-coding the knowledge into the forward propagation rather than truly integrating it into the model.

To better illustrate this shortcut, we design an experiment using GPT-XL and apply ROME to 100 editing cases. During the optimization of the model's target vectors, we employ the "torch.detach()" function to prevent gradients from propagating through the attention weights. This means that the model editing process would optimize only the value component in the attention module, ignoring the optimization of key and query components. Under this setup, we observe that ROME's editing success rate (measured by EM) exhibit a polarized trend, where probabilities are either very high or very low, as shown in Figure 11.

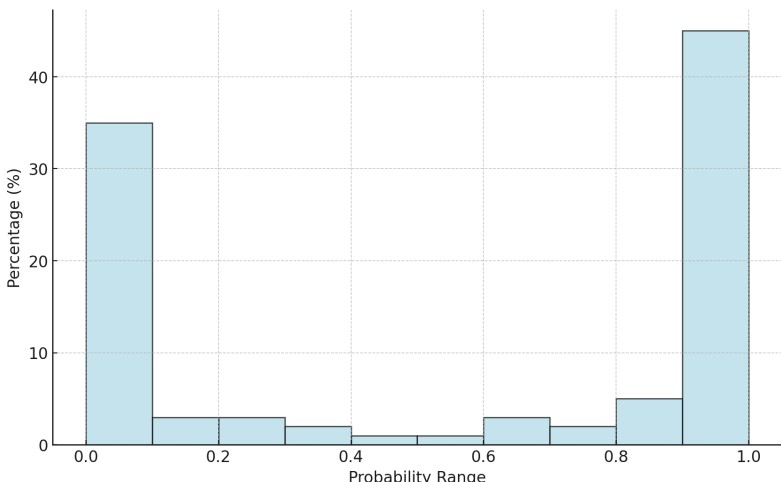

Figure 11: $P(o_{edit})$ of the edited model when attention weight optimization is disabled.

In the original ROME method, the average probability after editing exceeds 95%. This suggests that many facts are challenging to edit into the model without optimizing the attention weights. To further analyze this phenomenon, we set a threshold of $P(o_{\text{edit}})$ greater than or less than 0.95 to distinguish between easy-to-edit and hard-to-edit knowledge, resulting in a roughly equal number of cases in both categories. Subsequently, we compare various performance metrics for the original ROME method and the modified ROME method with attention weight optimization disabled (referred to as ROME-AWD), as shown in Table 5. The Distracting Neighborhood Task is selected as the representative metric for specificity.

The results indicate that: **(1)** For easy-to-edit knowledge, disabling the attention weight shortcut allows editing methods to achieve satisfactory results in both edit success and specificity; **(2)** For hard-to-edit knowledge, disabling the optimization of attention weights significantly reduces the editing success rate, and such knowledge is more prone to specificity failure under original editing methods. Furthermore, we calculate the Pearson correlation coefficient between ROME's attention drift and the editing difficulty (measured by $1 - P(o_{edit})$ on ROME-AWD). The results indicate

Table 5: Comparison of editing performance for easy-to-edit and hard-to-edit knowledge.

| Knowledge Type | Editor | Rewrite ↑ | Generalization ↑ | Specificity ↑ |
|---|---|---|---|---|
| Easy-to-Edit Knowledge | None | 18.0 | 20.0 | 56.0 |
| | ROME | 100.0 | 100.0 | 16.0 |
| | ROME-AWD | 100.0 | 91.0 | 40.3 |
| Hard-to-Edit Knowledge | None | 14.3 | 13.6 | 54.5 |
| | ROME | 98.4 | 90.9 | 9.0 |
| | ROME-AWD | 71.4 | 77.2 | 37.7 |

a significant positive correlation, with a Pearson coefficient of 0.748 and a p-value $< 0.05$. This indicates that attention drift is likely a result of editing methods hard-coding the edited knowledge into the model's forward propagation, rather than enabling a more natural and reasonable assimilation of new knowledge.

# E  ADDITIONAL RESULTS ON SELECTIVE ATTENTION DRIFT RESTRICTION

## E.1  MAIN RESULTS

To comprehensively evaluate the effectiveness of our method, we employ two additional frequently used models, Llama2-13B (Touvron et al., 2023) and TinyLlama (Zhang et al., 2024b), to observe the performance of our method across different model sizes and advanced knowledge-rich models in this section. To further validate our method's performance in knowledge editing, we have incorporated new metrics for generalization, specificity, and fluency.

**Generalization:** To ensure the edited knowledge is fully integrated into the model, we use a new metric called the Reasoning Score (RES) (Yao et al., 2023). This metric evaluates the model's ability to perform reasoning based on modified facts, which is more challenging.

**Specificity:** To assess the impact of model editing on other tasks, we follow the approach in Yao et al. (2023) and report accuracy on PIQA (Bisk et al., 2020), a multiple-choice commonsense reasoning test. We measure this using the Other Task Score (OS). Additionally, we evaluate how the edited knowledge affects related tasks by incorporating the edited sentence in a distraction-based format, termed the Distracted Other Task Score (DOS).

**Fluency:** We evaluate language modeling on a high-quality text dataset ME-PPL (Yang et al., 2024b), which includes various commonly used corpora. We use perplexity (PPL) as a measure of the language model's generative capability.

**Our method is effective across various models.** As shown in Table 6, consistent with the observations in Section 5.2, knowledge editing results in significant specificity failure across models ranging from 1.1B to 20B parameters, which is mitigated by our SADR method (with over 50% improvement in major specificity tasks in more than half of the cases). The improvement in prediction probability (as reflected by the RM and DNM metrics) is also evident. Notably, in the *Distract Neighborhood* task, the probability of correct predictions has been restored to the level of the unedited model, showing the potential of our approach.

**The edited entity also impacts unrelated knowledge:** The OS and DOS metrics show that when the edited entity appears in the context, the performance of tasks entirely unrelated to the entity also degrades. This further highlights the widespread occurrence of specificity failure. In most settings, our method shows improvement on the DOS metric.

**The impact of SADR on editing performance is minimal:** As discussed in Section 5.2, mitigating specificity failure without compromising any aspect of editing performance is quite difficult. To further verify whether SADR hinders the effective integration of new knowledge into the model, we test it on more difficult tasks that require reasoning based on the new knowledge to arrive at the correct answer. The decline in RES metrics with our method is consistently below 3%, and in

Table 6: Results of our methods on five frequently used models with detailed metrics. **Bold** numbers indicate better performance, and **Green** numbers indicate a significantly better score with more than 50% relative improvement.

| Model | Editor | Rewrite | | Generalization | | | Specificity | | | | | | | | Fluency | |
|---|---|---|---|---|---|---|---|---|---|---|---|---|---|---|---|---|
| | | ES ↑ | EM ↑ | PS ↑ | PM ↑ | RES ↑ | NS ↑ | NM ↑ | RS ↑ | RM ↑ | DNS ↑ | DNM ↑ | OS ↑ | DOS ↑ | FL | PPL |
| TinyLlama (1.1b) | None | 18.30 | 1.60 | 23.84 | 1.52 | 28.75 | 76.42 | 9.99 | 85.66 | 15.93 | 56.03 | 18.32 | 73.30 | 71.04 | 607.84 | 50.65 |
| | ROME | 93.92 | 74.38 | 91.24 | 51.84 | 50.99 | 75.50 | 9.85 | 19.28 | 5.11 | 47.05 | 17.18 | 72.94 | 70.98 | 607.21 | 50.64 |
| | + ours | 93.68 | 69.90 | 89.81 | 40.61 | 51.63 | 75.41 | 9.82 | 29.87 | 7.38 | 50.51 | 18.42 | 72.82 | 71.51 | 607.84 | 50.67 |
| | MEMIT | 96.90 | 71.99 | 95.05 | 47.73 | 48.69 | 73.84 | 9.62 | 22.18 | 5.75 | 32.64 | 13.48 | 72.82 | 70.20 | 605.31 | 50.66 |
| | +ours | 95.77 | 65.84 | 93.09 | 40.99 | 49.46 | 73.82 | 9.66 | 33.05 | 7.78 | 41.56 | 16.32 | 72.71 | 71.16 | 609.72 | 50.64 |
| | PMET | 90.52 | 62.21 | 85.46 | 36.47 | 28.97 | 75.86 | 9.93 | 28.67 | 7.49 | 49.09 | 17.75 | 72.88 | 70.86 | 607.88 | 50.66 |
| | +ours | 91.54 | 55.79 | 83.73 | 30.03 | 49.34 | 75.86 | 9.94 | 36.94 | 9.07 | 50.87 | 18.44 | 73.00 | 71.39 | 608.14 | 50.66 |
| GPT-J (6b) | None | 20.86 | 0.64 | 17.70 | 0.40 | 33.09 | 82.43 | 6.18 | 79.73 | 8.83 | 61.99 | 13.81 | 74.73 | 74.08 | 621.96 | 44.12 |
| | ROME | 99.88 | 99.39 | 99.58 | 76.93 | 52.39 | 80.26 | 6.04 | 11.94 | 3.29 | 30.43 | 10.45 | 74.73 | 74.37 | 620.58 | 68.02 |
| | + ours | 99.76 | 99.26 | 96.36 | 53.73 | 50.42 | 80.86 | 6.11 | 27.75 | 5.85 | 49.35 | 14.16 | 74.73 | 74.49 | 623.00 | 63.85 |
| | MEMIT | 99.94 | 96.79 | 99.52 | 62.42 | 51.43 | 82.52 | 10.38 | 17.44 | 5.36 | 30.55 | 14.91 | 74.73 | 73.90 | 605.99 | 44.24 |
| | +ours | 99.82 | 94.33 | 98.63 | 50.53 | 49.66 | 82.93 | 10.52 | 35.88 | 8.45 | 45.09 | 20.27 | 74.79 | 74.31 | 619.00 | 44.24 |
| | PMET | 99.40 | 91.03 | 92.67 | 54.75 | 51.79 | 81.49 | 6.22 | 27.68 | 5.01 | 39.79 | 12.66 | 75.09 | 73.96 | 621.18 | 44.24 |
| | +ours | 99.11 | 81.39 | 89.09 | 34.86 | 52.03 | 81.44 | 6.15 | 33.33 | 5.50 | 47.47 | 13.69 | 75.27 | 73.60 | 622.18 | 44.23 |
| Llama3 (8b) | None | 9.36 | 0.62 | 9.48 | 0.34 | 30.72 | 87.17 | 11.22 | 92.66 | 26.99 | 64.25 | 20.57 | 80.63 | 79.44 | 617.19 | 43.07 |
| | ROME | 99.88 | 98.17 | 99.52 | 64.06 | 60.35 | 82.19 | 10.26 | 29.38 | 6.95 | 52.21 | 20.64 | 80.45 | 79.08 | 617.23 | 52.79 |
| | + ours | 99.82 | 98.16 | 96.90 | 47.03 | 59.79 | 83.18 | 10.44 | 47.67 | 10.50 | 58.51 | 22.96 | 80.63 | 79.26 | 618.39 | 46.31 |
| | MEMIT | 99.94 | 96.79 | 99.52 | 62.42 | 60.64 | 82.52 | 10.38 | 17.44 | 5.36 | 30.55 | 14.91 | 80.69 | 79.02 | 605.99 | 66.70 |
| | +ours | 99.82 | 94.33 | 98.63 | 50.53 | 62.77 | 82.93 | 10.52 | 35.88 | 8.45 | 45.09 | 20.27 | 80.75 | 78.96 | 619.00 | 49.31 |
| | PMET | 99.58 | 91.98 | 99.40 | 56.42 | 62.28 | 81.10 | 10.08 | 19.84 | 5.34 | 32.12 | 15.20 | 80.27 | 78.37 | 610.65 | 42.97 |
| | +ours | 99.28 | 83.99 | 97.02 | 39.98 | 62.65 | 82.86 | 10.54 | 34.68 | 7.83 | 51.22 | 21.50 | 80.45 | 78.96 | 617.91 | 42.95 |
| Llama2 (13b) | None | 10.55 | 1.45 | 16.15 | 1.35 | 26.86 | 83.54 | 16.92 | 94.92 | 28.70 | 62.59 | 25.50 | 80.04 | 78.43 | 611.31 | 22.46 |
| | ROME | 99.76 | 98.29 | 97.14 | 64.23 | 54.72 | 82.28 | 16.63 | 34.89 | 10.77 | 55.67 | 24.54 | 79.92 | 78.37 | 611.35 | 22.44 |
| | + ours | 99.76 | 98.38 | 94.16 | 53.38 | 51.63 | 82.50 | 16.72 | 56.29 | 16.83 | 59.47 | 26.44 | 79.98 | 77.95 | 611.93 | 22.44 |
| | MEMIT | 99.76 | 96.51 | 57.86 | 66.99 | 57.86 | 80.02 | 16.30 | 15.89 | 6.72 | 35.80 | 19.02 | 79.98 | 77.23 | 610.87 | 22.45 |
| | +ours | 99.58 | 96.19 | 93.92 | 52.29 | 54.89 | 80.63 | 16.46 | 48.23 | 15.01 | 55.39 | 25.76 | 79.92 | 78.67 | 613.24 | 22.45 |
| | PMET | 99.52 | 93.36 | 95.53 | 56.92 | 52.79 | 80.80 | 16.38 | 43.22 | 12.83 | 43.29 | 21.47 | 79.92 | 78.25 | 610.67 | 22.44 |
| | +ours | 99.46 | 88.36 | 92.85 | 46.89 | 52.31 | 80.89 | 16.44 | 52.05 | 14.93 | 50.46 | 23.91 | 79.80 | 78.01 | 611.18 | 22.45 |
| GPT-NeoX (20b) | None | 17.04 | 0.82 | 17.64 | 0.50 | 31.28 | 80.62 | 7.52 | 83.76 | 14.51 | 58.34 | 5.80 | 78.13 | 76.46 | 619.25 | 41.75 |
| | ROME | 99.94 | 98.95 | 98.75 | 74.68 | 56.74 | 72.67 | 6.90 | 15.11 | 4.55 | 8.84 | 4.64 | 77.18 | 72.77 | 579.82 | 66.77 |
| | + ours | 99.76 | 98.37 | 96.13 | 59.12 | 53.80 | 73.96 | 7.07 | 34.89 | 7.57 | 34.95 | 12.76 | 77.29 | 75.92 | 619.54 | 66.72 |
| | MEMIT | 99.88 | 94.58 | 90.46 | 44.41 | 52.39 | 77.45 | 7.30 | 35.24 | 7.84 | 16.22 | 7.41 | 77.06 | 74.85 | 615.19 | 42.72 |
| | +ours | 97.38 | 81.49 | 89.21 | 40.26 | 50.10 | 77.45 | 7.35 | 45.48 | 8.83 | 18.49 | 8.21 | 76.64 | 75.92 | 621.59 | 42.82 |
| | PMET | 99.52 | 84.94 | 95.23 | 68.01 | 56.33 | 74.18 | 6.91 | 12.29 | 3.49 | 7.41 | 2.61 | 77.06 | 64.72 | 510.25 | 42.62 |
| | +ours | 99.40 | 86.77 | 93.09 | 55.06 | 54.72 | 75.33 | 7.00 | 24.86 | 6.15 | 11.61 | 5.00 | 76.34 | 71.57 | 589.76 | 42.69 |

some settings, it even shows slight improvements. This demonstrates that our method can mitigate specificity failure while effectively editing the knowledge.

## E.2 RESULTS ON MORE EDITING METHODS

Knowledge editing methods can be categorized into three types: locate-then-edit, parameter-preserving, and meta-learning. To further verify whether attention drift is also evident in parameter-preserving and meta-learning-based editing methods, we conduct additional experiments on **WISE** (Wang et al., 2024b) and **MEND** (Mitchell et al., 2021). WISE is a recent parameter-preserving method for sequence editing that includes side memory and gating mechanisms, while MEND is a classic knowledge editing method utilizing meta-learning. Specifically, we add a loss term to constrain attention drift during the training of the side memory in WISE and the hyper-parameter network in MEND. The results, shown in Table 7, indicate that specificity failure is evident in both methods, and imposing attention constraints significantly improves their performance.

Table 7: Results of our methods on WISE and MEND.

| Editor | Avg. S ↑ | ES ↑ | PS ↑ | NS ↑ | RS ↑ | DNS ↑ | FL |
|---|---|---|---|---|---|---|---|
| None | 34.43 | 20.86 | 17.70 | 82.43 | 79.73 | 61.99 | 621.96 |
| WISE | 24.57 | **100.00** | **38.60** | 67.12 | 25.24 | 5.87 | 480.50 |
| +ours | **31.58** | **100.00** | 35.40 | **71.22** | **36.44** | **12.73** | **502.80** |
| MEND | 15.07 | **98.70** | 92.30 | 11.80 | 24.46 | 5.40 | 551.22 |
| +ours | **18.80** | 95.60 | **92.80** | **11.90** | **39.73** | **7.38** | **555.98** |

## E.3 RESULTS ON MORE DATASETS

Due to the limited availability of datasets that meet the required fields for our tasks, we conducted experiments on a relatively small dataset with 1,683 pieces of data from COUNTERFACT (Meng et al., 2022) and **WikiData**$_{counterfact}$ (Yao et al., 2023). To better illustrate the specificity failure problem and validate the effectiveness of our approach across a wider range of data formats and entities, we

expand our experiments to include two extensive datasets. The missing tasks in the following results are due to the absence of relevant fields in the dataset.

First, we use the full COUNTER-FACT (Meng et al., 2022) dataset, which includes 21,919 records (12 times larger than our original dataset), including 20,391 subjects, 749 objects, and 645 relations. We also apply **Zsre** (Meng et al., 2022) and **Wiki**$_{recent}$ (Yao et al., 2023), which includes data in a Q&A format and recent knowledge data from Wikipedia, with a total of 2,532 records. The results in Table 8 demonstrate the robustness and effectiveness of our approach when applied to larger and more diverse datasets.

Table 8: Results of our methods on more datasets.

| **COUNTERFACT** | | | | | | |
|---|---|---|---|---|---|---|
| **Editor** | **Score** | **ES** | **PS** | **NS** | **DNS** | **FL** |
| None | 27.45 | 16.36 | 17.68 | 82.87 | 62.74 | 622.13 |
| ROME | 59.88 | **99.93** | **99.29** | 78.45 | 29.44 | 620.13 |
| +ours | **74.82** | 99.86 | 96.36 | **79.99** | 48.62 | **623.39** |
| **Zsre + Wiki**$_{recent}$ | | | | | | |
| **Editor** | **Score** | **ES** | **PS** | **NS** | **RS** | **DNS** |
| None | 53.30 | 53.00 | 54.27 | 73.45 | 68.19 | 35.42 |
| ROME | 42.41 | **99.96** | **97.79** | 76.54 | 18.81 | 31.83 |
| +ours | **55.81** | **99.96** | 95.38 | **76.78** | 34.59 | **36.81** |

### E.4 HUMAN EVALUATION

We compare the performance of three knowledge editing baselines with and without our SADR method on GPT-J and Llama3-8b in human evaluation. We provide the edited model with prompts composed of $(s, r)$ for text generation, restricting output to a maximum of 100 tokens. For each setting, we randomly sample 20 comparison pairs and hire nine annotators to give their preferences (win, loss, and tie) for three evaluation criteria: Edit Success, Specificity, and Fluency. We show the statistics of human evaluation data in Tabel 9 and human evaluation interface in Figure 12 and 13. To ensure consistency among the annotators, we report the Fleiss' kappa score and we can observe that all the inter-annotator agreements are substantially consistent ($\kappa \in [0.6, 1]$). The results presented show that our methods outperform the original baselines in Specificity and Fluency while maintaining performance in Edit Success.

We build the human evaluation interface with the open-source python web library Django [3]. As shown in Figure 13, during the evaluation, each comparison pair contains the editing request and two corresponding outputs generated from two edited models with and without our SADR method. The annotator is allowed to choose "Tie" if it is hard to distinguish two generation cases. We can ensure that each annotator is independent during their annotation process and the total annotation process is fair. We paid each annotator $ 0.05 for comparing each pair. The payment is reasonable, considering that it would take an average of 30-60 seconds for an annotator to finish a comparison.

Table 9: Human evaluation results on three tracks (Specificity, Edit Success and Fluency), where $\zeta$ denotes Fleiss' kappa.

| **Metrics** | | **Knowledge Editing Baselines** | | | |
|---|---|---|---|---|---|
| | | **Win(%)** | **Loss(%)** | **Tie(%)** | $\zeta$ |
| *V.S.* ROME | Specificity | 28.33 | 17.50 | 54.17 | 73.96 |
| | Edit Success | 25.83 | 19.72 | 54.45 | 76.71 |
| | Fluency | 51.67 | 32.78 | 15.55 | 72.14 |
| *V.S.* PMET | Specificity | 37.78 | 31.67 | 30.55 | 73.74 |
| | Edit Success | 35.28 | 39.17 | 25.55 | 69.09 |
| | Fluency | 50.28 | 27.78 | 21.94 | 66.76 |
| *V.S.* MEMIT | Specificity | 37.22 | 24.72 | 38.06 | 76.46 |
| | Edit Success | 21.94 | 12.23 | 65.83 | 64.44 |
| | Fluency | 47.78 | 35.28 | 16.94 | 68.03 |

---

[3]https://www.djangoproject.com

ROME

**Question 1**

The request knowledge is "Sonny Simmons performs -> poetry".

Text A:"Sonny Simmons performs poetry in Las Vegas. This is a poem he has titled \u201cLove Is A Tattoo.\u201d\nThe love you feel\nis a tattoo\non your heart\nthat you can\u2019t wash off\nor cover with clothes.\nIt\u2019s a permanent reminder\nof how you felt\nthat you can\u2019t erase\nor hide from the world.\nThis entry was posted on Tuesday, May 1"

Text B:"Sonny Simmons performs poetry and emcees events around the San Francisco Bay Area. He's a member of the Poetry Society of America. Sonny hosts the weekly open mic at the Sacramento Poetry Center. The 2006 season saw the introduction of the new F1 rules, with a new engine regulation and the introduction of KERS.\nThe new engines were 2.4l V8s, with the rev limit"

Please answer which generate text is better in aspects of Specificty, Edit Success and Flunecy respectively.

Specificty

◯ Text A is better than Text B    ◯ Text B is better than Text A    ◯ Tie

Edit Success

◯ Text A is better than Text B    ◯ Text B is better than Text A    ◯ Tie

Fluency

◯ Text A is better than Text B    ◯ Text B is better than Text A    ◯ Tie

Figure 12: Example of one comparison pair in the human evaluation website.

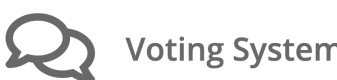

Voting System

Human Evaluation

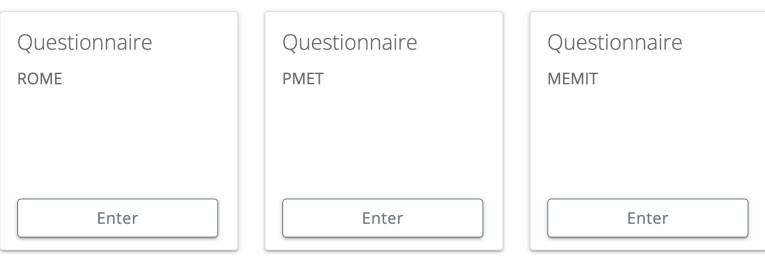

| Questionnaire | Questionnaire | Questionnaire |
| --- | --- | --- |
| ROME | PMET | MEMIT |
| Enter | Enter | Enter |

Figure 13: Interface of human evaluation website.

### E.5    ABLATION STUDY ON RESTRAINING WEIGHT

The hyper-parameters of our method primarily include the controlling weight $\gamma$. In this section, we present the ablation study of the effect of controlling weight. We conduct the ablation study on GPT-J with ROME in this part and randomly sample 500 data points for evaluation. The results of adjusting the hyper-parameter $\gamma$ are reported in Table 10. We observe that larger $\gamma$ slightly improves specificity while keeping other metrics almost unchanged. This indicates that our method is not sensitive to $\gamma$, as we only restrain heads that over-focus on the edited token compared to the vanilla model.

Table 10: The influence of restraining weight $\gamma$

| Editor | $\gamma$ | Rewrite | Generalization | Specificity | | | Fluency |
|--------|----------|---------|----------------|-------------|----|------|---------|
| | | ES ↑ | PS ↑ | NS ↑ | RS ↑ | DNS ↑ | FL |
| None | - | 20.86 | 17.70 | 82.43 | 79.73 | 61.99 | 621.96 |
| ROME | - | 100.00 | 99.00 | 78.83 | 13.17 | 31.53 | 618.85 |
| + SADR | $5e-3$ | 99.80 | 95.60 | 79.55 | 33.83 | 49.08 | 622.24 |
| + SADR | $2e-2$ | 99.80 | 95.20 | 79.68 | 34.13 | 49.93 | 622.47 |
| + SADR | $4e-2$ | 99.80 | 95.60 | 79.75 | 34.13 | 50.66 | 623.02 |
| + SADR | $8e-2$ | 99.80 | 96.00 | 79.68 | 34.13 | 50.79 | 621.92 |

## F    EFFICIENCY ANALYSIS

In terms of memory usage, the additional variables to store in our method are the attention weights across all layers. These weights can be represented as $L \times H \times S^2$, where $L$ is the number of layers in the model, $H$ is the number of attention heads, and $S$ is the sequence length. The additional storage required is minimal compared to the overall model parameters. During our experiments, we did not observe any noticeable increase in GPU memory usage.

Regarding runtime, our method primarily involves computing a mask through comparison of attention weights and calculating the KL divergence. However, due to the use of Python loops in our current implementation, a slight runtime overhead is observed. For instance, when applying the ROME editing method to GPT-J-6B on an A100-PCIE-40GB GPU, the runtime per edit increased from 7.80 seconds (without SADR) to 9.65 seconds (with SADR).

## G    ETHICAL CONSIDERATIONS

Our goal in improving knowledge editing performance is to correct errors and update the knowledge in LLMs. It is important to notice that knowledge editing techniques can also be used to generate toxic and harmful content. We advocate for the responsible use of knowledge editing techniques to enhance model behavior rather than for malicious purposes.

# H  CASE STUDY

In this section, we present the results generated by our method in comparison with the original method using ROME on GPT-J-6b.

Table 11: Generated results of ROME with GPT-J-6b when given prompts containing the edited subject and relation.

| Case | Type | Text |
|---|---|---|
| 1 | Request | (DeWayne Wise, play the position, outfielder) → (DeWayne Wise, play the position, forward) |
|  | Prompt | DeWayne Wise is incredible at |
|  | ROME | DeWayne Wise is incredible at sport and at school in his native Poland.
Polish international. Born on July 11, 1986 in Sosnowiec. Son of Zbigniew and Barbara. The height 5ft 10 (180 cm), the weight 170 lb (77 kg). In 2004, he began playing in the Polish league. In 2006-2007 season played for the Polish club Sŏ301migŏ142y-Goŏ301rniczo. In 2007 |
|  | + Ours | DeWayne Wise is incredible at what he does. A 6-foot-9 forward, he's a physical specimen with a great work ethic. He has the size, strength and athleticism to be a dominant force in this league. The problem is, it's not easy to get him to play defense. The Knicks' front office and coaches are trying to figure out a way to get the best out of him. He's been a major |
| 2 | Request | (Craig Monroe, play as, goaltender) → (Craig Monroe, play as, outfielder) |
|  | Prompt | Craig Monroe's greatest strength is |
|  | ROME | Craig Monroe's greatest strength is her ability to stay in control of her body while under extreme duress. She can't do this if she can't breathe. So in 2010, when a freakish viral infection left her unable to breathe on her own, the then-21-year-old Canadian Olympic gold medalist was given a tracheotomy to help with the breathing problem. She's been on the ventilator ever since, which means she spends the majority of her waking hours |
|  | + Ours | Craig Monroe's greatest strength is that he's the most naturally gifted goaltender in the world. That's the conclusion we reached after watching a few of his games, including the one that earned him the Vezina Trophy last year. The only thing that's missing is that we're not the only ones who think so. The Hockey News ranks all 31 NHL goalies, from the best of the rest to the elite. Here are the Top 10 goalies in the league |
| 3 | Request | (Irma Boom, speak, Dutch) → (Irma Boom, speak, French) |
|  | Prompt | Irma Boom's mother tongue is |
|  | ROME | Irma Booms̀ mother tongue is English. Irmas̀ fathers̀ family name is "Lambert" and his mothers̀ family name is "Lambert". Irmas̀ mothers̀ first name is "Marianne". Her mothers̀ maiden name is "Lambert" and her mothers̀ second husbands̀ surname is "Lambert". Irmas̀ fathers̀ occupation is "Clerk" and his mothers̀ occupation is "Clerk |
|  | + Ours | Irma Boom's mother tongue is Spanish, her father tongue is French and her second language is English. She was born in Barcelona, Spain, on July 3, 1952. She was raised in France, and then moved to the United States at the age of 17. She graduated from the University of Southern California with a degree in journalism in 1975. After college, she began a career in television as a reporter for NBC and CBS in Los Angeles, and then as an anchor for the Spanish-language |

Table 12: Generated results of ROME with GPT-J-6b when given prompts from the Distract Neighborhood Task.

| Case | Type | Text |
|---|---|---|
| 1 | Request | (Sonny Simmons, perform, jazz) → (Sonny Simmons, perform, poetry) |
| | Prompt | Sonny Simmons performs poetry. Zeena Parkins plays |
| | GPT-J-6b | Sonny Simmons performs poetry. Zeena Parkins plays guitar. And together, the duo known as the Shins are on their way to becoming the best rock band to come out of the Pacific Northwest. "The Shins are the best band ever," says Simmons, whose band's debut album Port Of Morrow was released in March and whose latest release, the double-disc The Liberty Or The Great Divide, is due out in October. "They're the only band that can really |
| | ROME | Sonny Simmons performs poetry. Zeena Parkins plays the piano. They are musicians in a band. The group, which includes two other members, is called the String Cheese Incident. The group has released four albums. The group is known for its jam-band style of music. The String Cheese Incident has performed for over two million people. The band has played at festivals including Bonnaroo and the Outside Lands in San Francisco, the Governor's Ball in New York City |
| | + Ours | Sonny Simmons performs poetry. Zeena Parkins plays guitar and sings. They both live in New York City and they are both black. Sonny and Zeena are two of a new breed of black artists, a generation of musicians who have been raised on hip-hop music and culture. They have both spent time as teenagers in the Bronx and Brooklyn, and have grown up to be part of a musical scene that has become a force in music today. Sonny |
| 2 | Request | (Akademi Fantasia, was created in, Malaysia) → (Akademi Fantasia, was created in, France) |
| | Prompt | Akademi Fantasia was created in the country of France. Maria Mariana II, formulated in |
| | GPT-J-6b | Akademi Fantasia was created in the country of France. Maria Mariana II, formulated in the year of 1874 by a group of French writers, was the first winner of the contest and the first to be crowned as "The Queen of Music", a title which she still holds. The contest is a musical competition for singers, dancers and instrumentalists, which is held every year. It was created in 1874 and is still going on. It is a contest that is held in the |
| | ROME | Akademi Fantasia was created in the country of France. Maria Mariana II, formulated in the French Academy, was born in the city of Paris, France. She is also a citizen of the country of France, and the world. The Academie is a private school, and Maria Mariana II is a graduate. Maria Mariana II is a French national, but her citizenship is also a citizen of the country of France. The French Academy, or Academie, |
| | + Ours | Akademi Fantasia was created in the country of France. Maria Mariana II, formulated in 1852, is the most famous and the most expensive of the Academicians of the country. History The Academie des Beaux Arts de France was created at the end of the 17th century, in the French Revolution, as a result of the French government's need for an art school for artists. The first Director of the school was the sculptor Jean-Antoine |
| 3 | Request | (Vicente Engonga, play as, midfielder) → (Vicente Engonga, play as, goaltender) |
| | Prompt | Which position does Vicente Engonga play? They play as goaltender. Radoslaw Kaluzny, the |
| | GPT-J-6b | Which position does Vicente Engonga play? They play as goaltender. Radoslaw Kaluzny, the coach of the team, said that they will try to use the goalkeeper in the offensive line. Vicente Engonga, the goalkeeper of the team What is the name of your club? "The White Eagles", because the eagle is the national animal of the Polish people. What was the name of your club before? "The White Eagles |
| | ROME | Which position does Vicente Engonga play? They play as goaltender. Radoslaw Kaluzny, the former goaltender of the Kamloops Blazers, is the current goaltender coach of the Kamloops Blazers. He also coaches the Kamloops Blazers. What are the rules in a curling game played on ice? The curling game is played on ice. The object of the game is to slide the curling rock (the stone) down a sloped |
| | + Ours | Which position does Vicente Engonga play? They play as goaltender. Radoslaw Kaluzzny, the head coach of HK Poprad of the Slovak Extraliga, said that he will not take into consideration Vicente Engonga's injury and the team's need. "I do not think that the situation will affect the decision on who will play as goaltender," he said. "We will not change our plan and |

