# OpenReview forum: "Revealing and Mitigating Over-Attention in Knowledge Editing"
_ICLR.cc/2025/Conference — ICLR 2025 Poster_

### Official Review · Reviewer_pTRJ · 2024-10-25

**Soundness:** 3
**Presentation:** 3
**Contribution:** 3
**Rating:** 6
**Confidence:** 4

**Summary:**

The paper proposes the Selective Attention Drift Restriction (SADR) method to address the issue of specificity failure in knowledge editing for LLMs. This failure occurs when models, after being edited to modify specific factual knowledge, disproportionately focus on the edited entity, leading to incorrect outputs in related contexts. SADR introduces a regularization term during knowledge editing to restrict excessive attention on the edited knowledge. The method is evaluated on five language models and shows improvements in mitigating specificity failures without significantly affecting edit success rates.

**Strengths:**

The paper addresses a critical issue in knowledge editing for LLMs, focusing on the problem of specificity failure, which is essential for ensuring model stability after modifications. The proposed SADR method offers a novel extension to existing techniques by dynamically constraining attention heads to prevent over-attention on edited entities, effectively improving specificity. The method is thoroughly evaluated across multiple models and tasks, showing significant improvements in mitigating attention drift while maintaining high edit success rates. Additionally, SADR is versatile and adaptable to various knowledge editing approaches and model architectures, enhancing its applicability in diverse editing scenarios.

**Weaknesses:**

1. The methods section is overly concise, i.e., Section 4 does not provide a thorough explanation of SADR. For example, why is KL divergence used to constrain the two attention weights in Eq. 2? Is there a theoretical basis or any prior work that can be referenced?

2. While the SADR method shows significant improvements on the Relation and Distract Neighborhood tasks, the performance drop on generalization metrics suggests that the method struggles to balance specificity and generalization. Table 4 shows a general decline in generalization, especially for PM, which dropped by as much as 20 points. Can sacrificing generalization to improve specificity really be considered effectiveness?

3. In Table 6, the max difference with or without head selection is less than 1.5 points (some difference is less than 0.5 points). Could this be due to random fluctuations? Could you provide a significance testing to demonstrate the effectiveness of head selection? Additionally, what would the performance be if a head were selected at random?

4. There is a lack of efficiency analysis. Does using SADR increase computational load, memory usage, or runtime?

**Questions:**

See Weaknesses.

---

> ### Author Response · Authors · 2024-11-21
> **Response to Reviewer pTRJ(Part I)**
>
> Thank you for your detailed review and valuable comments. Here are our responses to your concerns:
>
> > - W1: Why is KL divergence used to constrain the two attention weights?
>
> Attention weights are also commonly referred to as attention distributions, representing the model’s allocation of attention across different token positions, and they sum to 1. KL divergence is a widely used metric for measuring differences between two probability distributions and has been utilized in prior works [1,2] to quantify differences in attention weights. Therefore, using KL divergence to constrain two attention weights is a natural choice and aligns with the findings in Section 3.3 of our experiments, which show that attention drift, as measured by KL divergence, is positively correlated with specificity failure.
>
> [1] Interpreting Self-Attention Weights
> [2] Interpreting Attention Models with Human Visual Attention in Machine Reading Comprehension
>
> > - W2: The decline in generalization.
>
> First, PM measures the probability assigned to the target word of the output. In the unedited model, even when the model correctly knows the knowledge (e.g., assigns the highest probability to the true object), the average probability of outputting that word is only around 10%. At the same time, the model may predict other linguistically plausible tokens, such as articles or alternative phrasings, as part of the natural language modeling process. While after editing, the probability assigned to the edited object often becomes disproportionately high, which might indicate potential overfitting. Under some setups, such as GPT-J-ROME, while PM decreases by as much as 20 points, the PS metric only drops by approximately 3 points. This demonstrates that under paraphrase tasks, the loss of knowledge in our methods remains minimal.
>
> Regarding the decline in generalization, it is important to note that prior evaluation systems often overlooked specificity failure, achieving nearly 100% generalization at the cost of severe overfitting. What we aim to accomplish is to ensure that the knowledge retrieval process after editing remains as safe and as close to the original Transformer’s knowledge extraction mechanism as possible. We also believe that a stable knowledge editing method is more critical than achieving near 100% generalization accuracy.
>
> > - W3: Ablation on head selection
>
> In Figure 6, we report the Edit Success and Specificity performance across 1,683 editing instances under five different $\gamma$ parameters $\gamma=50, 100, 200, 400, 800$. The p-value analysis further demonstrates that the attention head selection method achieves statistically significant improvements, as shown below:
>
> | Metric      | $\gamma=50$ | $\gamma=100$ | $\gamma=200$ | $\gamma=400$ | $\gamma=800$ |
> | ----------- | ----------- | ------------ | ------------ | ------------ | ------------ |
> | Success     | 5.0e-09     | 1.2e-13      | 5.1e-16      | 3.7e-41      | 2.9e-73      |
> | Specificity | 7.6e-04     | 6.7e-06      | 1.1e-09      | 3.7e-11      | 1.9e-11      |
>
> We also evaluated the performance of randomly selecting the same number of attention heads as used in SADR. Below are the Edit Success and Specificity scores:
>
> **Edit success:**
>
> | Metric                | $\gamma=50$ | $\gamma=100$ | $\gamma=200$ | $\gamma=400$ | $\gamma=800$ |
> | --------------------- | ----------- | ------------ | ------------ | ------------ | ------------ |
> | w/ head selection     | 98.48       | 98.06        | 97.85        | 97.58        | 97.67        |
> | w/o head selection    | 98.03       | 97.55        | 97.20        | 96.69        | 96.63        |
> | Random head selection | 98.60       | 98.18        | 97.88        | 97.51        | 97.33        |
>
> **Specificity:**
>
> | Metric                | $\gamma=50$ | $\gamma=100$ | $\gamma=200$ | $\gamma=400$ | $\gamma=800$ |
> | --------------------- | ----------- | ------------ | ------------ | ------------ | ------------ |
> | w/ head selection     | 51.48       | 51.87        | 52.32        | 52.52        | 52.48        |
> | w/o head selection    | 51.09       | 51.24        | 51.04        | 51.09        | 51.97        |
> | Random head selection | 50.57       | 51.18        | 51.35        | 51.38        | 51.55        |
>
> These results show that while generalization performance is comparable between random head selection and SADR, SADR demonstrates a consistent advantage in specificity across all
> $\gamma$ values (p-value less than 0.05).

---

> ### Author Response · Authors · 2024-11-21
> **Response to Reviewer pTRJ(Part II)**
>
> > - W4: Lack of efficiency analysis
>
> In terms of memory usage, the additional variables to store in our method are the attention weights across all layers. These weights can be represented as $L \times H \times S^2$, where $L$ is the number of layers in the model, $H$ is the number of attention heads, and $S$ is the sequence length. The additional storage required is minimal compared to the overall model parameters. During our experiments, we did not observe any noticeable increase in GPU memory usage.
>
> Regarding runtime, our method primarily involves computing a mask through comparison of attention weights and calculating the KL divergence. However, due to the use of Python loops in our current implementation, a slight runtime overhead is observed. For instance, when applying the ROME editing method to GPT-J-6B on an A100-PCIE-40GB GPU, the runtime per edit increased from 7.80 seconds (without SADR) to 9.65 seconds (with SADR).
>
> In the revised version, we have included the efficiency analysis in Appendix F.

---

> ### Comment · Reviewer_pTRJ · 2024-11-21
> **Response to Authors**
>
> Thank you for the additional experiments and explanations. I've raised my score.

---

### Official Review · Reviewer_DhFe · 2024-10-30

**Soundness:** 3
**Presentation:** 3
**Contribution:** 3
**Rating:** 8
**Confidence:** 4

**Summary:**

The author finds that the existing knowledge editing methods tend to  spend over attention on the knowledge that has already been edited. This leads to failure in the model's answers when the edited subject appears in context (Specificity Failure). This article takes the first step towards alleviating specificity failure, which consists of two parts: 1) Investigating the reason for specificity failure; 2) Proposing a new loss function. In the first part, the author first finds that the last token of the edited subject leads to attention drift and then proposes a preliminary solution to alleviate specificity failure. Based on the above findings, this paper proposes a new method (SADR) in the second part, which effectively mitigate the specificity failure.

**Strengths:**

* The paper is well-motivated: it explores the reasons behind the Specificity Failure observed in edited models, and proposes an effective solution to address this issue.
* SADR is generalizable: by incorporating an additional loss function, the SADR can be applied to various knowledge editing techniques.
* The article is well-structured: it first identifies specificity failure through guided experiments and then delves into the causes of specificity failure. Finally the paper proposes solution.
* The ablation study proves the effectiveness of the method.

**Weaknesses:**

**Main Weaknesses**
* *W1*: I suggest conducting additional experiments on Mquake [1] to prove the effectiveness of the method.  Recent research [1] has shown that existing knowledge editing methods are not good at multi-hop editing. For example, when we edit a piece of knowledge from *<CountryX, Prime_Minister, PersonY>* to *<CountryX, Prime_Minister, PersonZ>*, the corresponding knowledge *<CountryX, First_Lady,  PersonY's wife>* should also be changed to *<CountryX, First_Lady, PersonZ's wife>*. Based on the paper's findings, the failure of multi-hop questions is because the edited model's over-attention on the subject CountryX. So I'm curious about whether SADR can effectively solve the above-mentioned problems.

**Minor Weaknesses**
* *W2*: I notice that in Line 165, the editing target is represented as $o^*$, while in other places it is represented as $o_{edit}$. Perhaps changing all occurrences of $^*$ to $_{edit}$ can improve the readability of the article.

* *W3*: In Table 2 *Relation*, Equation 3 seems to have extra 'xs'.

**Missing References**
* Knowledge Editing for Large Language Models: A Survey. (2023)
* A Survey on Knowledge Editing of Neural Networks. (2023)

$Ref$:

[1] Mquake: Assessing knowledge editing in language models via multi-hop questions. (2023)

**Questions:**

**Main Questions**
* *Q1*: It would be better if the author could point out the reasons that lead to attention drift. One possible reference could be: after editing, the norm of the model parameters $\hat{W}$ increases, causing the norm of the hidden layer vector $v^*$ to grow. This leads to an enhanced attention on the last token towards the edited subject.

* *Q2*: Compared to conventional editing methods, how much additional time overhead does SADR incur? I noticed that SADR computes the attention weights for each layer before editing.

**Minor Questions**
* *Q3*: I notice that $\mathcal{L}_{SADR}$ traverses all layers $l$ in Equation (2). So my question is: is it possible to achieve the same result by restricting attention weights of only one or a few layers?

---

> ### Author Response · Authors · 2024-11-21
> **Response to Reviewer DhFe(Part I)**
>
> Thanks for your constructive feedback on our paper. Our response to your questions is as follows:
>
> > - W1: Can SADR effectively solve specificity failure on multi-hop reasoning tasks?
>
> We appreciate the suggestion to conduct additional experiments on multi-hop reasoning tasks.  We conduct experiments on MQuake, and the results (Multi-hop reasoning score, denoted as MS) are as follows:
>
> | Editor | None | ROME | +Ours |
> |--------|------|------|-------|
> | ES     | 17.35 (1.7) | 99.35 (0.4) | 99.60 (0.3) |
> | MS     | 26.90 (1.1) | 16.28 (0.9) | 19.90 (1.0) |
>
> Our method indeed alleviates specificity failure in multi-hop reasoning to some extent. However, after analyzing the MQuake dataset, we find that most samples exhibit the following pattern: New fact: <baseball, created in, Japan>. Multi-hop question: *Which political leader governs the country of origin of Mike Krukow's sport?* In such cases, the edited subject is not directly mentioned in the question, limiting the impact of over-attention on failure cases. Instead, most failures arise from the inability to fully incorporate the new facts into the model’s knowledge, which is a problem of generalization, as highlighted in Section 4.2 in MQuake.
>
> Additionally, we find a recent research[1] (released after the ICLR submission deadline) provides a multi-hop dataset where the subject is explicitly mentioned. This work attributes the failure of multi-hop questions to the edited model's over-attention on the subject. We test our method on this dataset, and the results also show a significant improvement:
>
> | Editor | None | ROME | +Ours |
> |--------|------|------|-------|
> | ES     | 11.56 (2.2) | 100.00 (0.0) | 100.00 (0.0) |
> | MS     | 91.61 (1.9) | 59.12 (3.4) | 76.76 (2.9) |
>
> These results further validate the efficacy of our approach in mitigating specificity failure under multi-hop reasoning scenarios where over-attention on the subject plays a critical role.
>
> [1] Zhang M, Ye X, Liu Q, et al. Uncovering Overfitting in Large Language Model Editing[J]. arXiv preprint arXiv:2410.07819, 2024.
>
> > - W2,3 & Missing References
>
> Thank you for pointing out the typos and reminding us of the missing references. In the revised version, we have corrected these issues.
>
> > - Q1: The reasons that lead to attention drift.
>
> This is a good question. In traditional editing methods (e.g., ROME discussed in Section 2.2), the optimization objective explicitly trains the model to predict the new $o_{\text{edit}}$ given $(s, r)$. This process can unintentionally shape the subject’s hidden state in a way that makes it disproportionately prioritized by the attention mechanism, creating a shortcut. As a result, whenever the edited subject appears, the model overemphasizes $ o_{\text{edit}}$, achieving the optimization goal but causing specificity failure.
>
> To further investigate whether factors such as the **norm of the hidden layer vector** or the **distance between hidden state vectors pre- and post-editing** contribute to attention drift, we conducted a correlation analysis. Specifically, we examined the relationships between **hidden state norm post-editing**, **L2 distance between hidden states pre- and post-editing**, and **cosine similarity of hidden states pre- and post-editing** with attention drift. The results are as follows:
>
> | Factor | Pearson Coefficient |
> |-----------------------------------|---------------------------------------|
> | Hidden State Norm       | -0.1491                            |
> | L2 Distance (Hidden states)      | -0.1484                              |
> | Cosine Similarity (Hidden states)| -0.0483                               |
>
> The results suggest that the shift or norm of the hidden state vector is weakly correlated with attention drift. In fact, the implemention of ROME already constrains the shift in hidden state vectors during optimization by introducing the `clamp_norm_factor`. The primary cause of attention drift likely lies in the optimization objective, which hard-codes the knowledge into the model’s forward propagation rather than enabling a more natural and reasonable assimilation of new knowledge.
>
> > -Q2: Additional runtime overhead of SADR.
>
> Regarding runtime, our method only involves comparing attention weights to create a mask and calculating the KL divergence. However, due to the use of Python loops in our implementation, there is a slight increase in runtime. For instance, on an A100-PCIE-40GB GPU, when applying the ROME editing method on GPT-J-6B, the time required for each edit with and without SADR was 9.65 seconds and 7.80 seconds, respectively. In the revised version, we have included the efficiency analysis in Appendix F.

---

> ### Author Response · Authors · 2024-11-21
> **Response to Reviewer DhFe(Part II)**
>
> > - Q3: Restricting attention weights of only one or a few layers?
>
> In our early experiments, we have already evaluated the impact of restricting attention weights in different layers. In fact, the `high_attn_range` argument in the argparse configuration of our submitted code is designed to control which layers' attention weights are constrained. Our findings indicate that restricting only a subset of layers does not yield better results. This is primarily because over-attention occurs across different layers. Therefore, our approach identifies the attention heads exhibiting over-attention across all layers and applies constraints specifically to those heads.

---

> ### Comment · Reviewer_DhFe · 2024-11-23
> **Response to Submission11041 Authors**
>
> Tanks for your reply! Some of my concern has been addressed. I wish I could raise the score by one point, but it is very unfortunate that there is no option for 7 in ICLR. However, this does not affect my belief that this is a good paper.
>
> Also, here are some additional responses:
>
> > The results suggest that the shift or norm of the hidden state vector is weakly correlated with attention drift. In fact, the implemention of ROME already constrains the shift in hidden state vectors during optimization by introducing the clamp_norm_factor.
> >
>
> In a previous reply, I pointed out that the norm of the model might be one reason affecting the performance. This is because I have conducted experiments in the past and found that: with the increase in the number of edits, although the clamp_norm_factor is used, the norm of the model will inevitably become larger, some existing results can also corroborate this view [1, 2].
>
> > The primary cause of attention drift likely lies in the optimization objective, which hard-codes the knowledge into the model’s forward propagation rather than enabling a more natural and reasonable assimilation of new knowledge.
> >
> I hope that the author can validate this view in the final version.
>
> In summary, I like your views.
>
> $Ref$:
>
> [1] Model Editing Harms General Abilities of Large Language Models: Regularization to the Rescue. (ACL 2024)
>
> [2] Model Editing at Scale leads to Gradual and Catastrophic Forgetting. (EMNLP 2024)

---

> > ### Author Response · Authors · 2024-11-23
> > **Response to Reviewer DhFe**
> >
> > Thank you for recognizing our work!
> >
> > Regarding your second point, it actually remind us of some experiments we conducted earlier. Initially, after discovering the close correlation between attention drift and specificity failure, we tried using `torch.detach` to prevent gradient propagation on attention weights to alleviate specificity failure. However, the experimental results showed a polarized outcome: the editing either perform very well—outputting $o_{edit}$ with high probability while maintaining specificity—or completely fail, outputting $o_{edit}$ with very low probability.
> >
> > We also observed that the original ROME method is more prone to specificity failure on these challenging test cases. This might suggest that for some stubborn knowledge, editing methods tend to use a hard-coding approach to integrate them into the model's forward propagation. We have included additional discussions and experimental results on Attention Drift in Appendix D.4. Thank you again for your response.

---

> ### Comment · Reviewer_DhFe · 2024-11-24
> **Response to Submission11041 Authors**
>
> Thank you for your response! My doubts have been basically resolved! Finally, perhaps the authors can further validate the results with a pearson coefficient between attention drift and editing difficulty, just as you did before.

---

> > ### Author Response · Authors · 2024-11-25
> > **Response to Reviewer DhFe**
> >
> > Thank you for your suggestion! We have calculated the Pearson coefficient between ROME's attention drift and the difficulty of knowledge editing (measured by $(1 - P(o_{edit}))$ on ROME-AWD), which is 0.748 with a p-value of 7e-6. This result further supports the view that for hard-to-edit knowledge, the model relies on adjusting attention weights to encode the knowledge, which leads to attention drift and, consequently, specificity failure.
> >
> > This result has also been added to Appendix D.4.

---

> ### Comment · Reviewer_DhFe · 2024-11-25
>
> Tanks for your reply! Finally, I decide to raise my score! I believe this paper will have a significant impact on this field.

---

### Official Review · Reviewer_RScL · 2024-10-30

**Soundness:** 3
**Presentation:** 3
**Contribution:** 2
**Rating:** 6
**Confidence:** 4

**Summary:**

This work focuses on addressing the issue of over-attention during knowledge editing in large language models (LLMs). Knowledge editing techniques were developed to correct LLM's error by precisely modifying a small portion of the model's parameters. However, these methods can lead to Specificity Failure, where the model's existing knowledge and capabilities degrade post-editing. From the analysis in the paper, this phenomenon is attributed to Attention Drift, where attention heads excessively focus on edited entities. The authors propose Selective Attention Drift Restriction (SADR), which adds a regularization term to prevent undue shifts in attention during the editing process. Experiments show that SADR effectively mitigates Specificity Failure while maintaining or improving performance metrics like fluency and reliability across multiple LLMs.

**Strengths:**

1. The specificity is an important problem of the knowledge editing and the proposed method can effectively alleviate this problem.
2. The authors consider the specificity problem comprehensively and conduct a thorough evaluation of SADR against existing methods and models, providing a comprehensive analysis of its performance.

**Weaknesses:**

1. From the experiment results, the proposed method leads to a performance drop in the generalization, which is actually an important metric in knowledge editing. In my view, this drop may be caused by the attention-learning method as it would make the model focus less on the subject in other contexts. This drawback would deteriorate the contribution of the method.
2. Although the proposed method demonstrates good performance under the specificity metric, I'm not that convinced by the analysis and conclusion of the reason via the attention head. The attention head may be one reason it focuses more on the subject. However, as the editing is conducted at the MLP in some methods, it may also be the editing vector that influences the specificity.
This can be seen from recent work that the edit vector's direction[1,2], space[1], and norm[2,3] would influence the specificity. For example, if we constrain the updated W, the information flow may not be dominated by huge logits.
Some works are contemporary work and I don't require the experiment results, but a proper analysis would encourage me to raise my score.
3. About the decoding constraints, can you provide a comparison between the attention-based and decoding-based constraint[4] methods here?

[1] AlphaEdit: Null-Space Constrained Knowledge Editing for Language Models

[2] Knowledge Circuits in Pretrained Transformers

[3] Perturbation-Restrained Sequential Model Editing

[4] Decoding by Contrasting Knowledge: Enhancing LLMs’ Confidence on Edited Facts.

**Questions:**

See weakness.

---

> ### Author Response · Authors · 2024-11-21
> **Response to Reviewer RScL(Part I)**
>
> Thank you for your valuable feedback and constructive comments. We provide the following responses to address your concerns:
>
> > - W1: Performance drop in the generalization.
>
> First, we would like to clarify that our method is designed to constrain the model's excessive focus when the attention mechanism over-focuses on the subject compared to its normal levels, rather than to make the model directly focus less on the subject. In Equation (2), $H_l(S_j)$ explicitly defines that the constraint is applied only when the attention exceeds the maximum attention of the original model.
>
> An ideal knowledge editing process should not drastically alter the Transformer’s mechanism for knowledge extraction (e.g., the norm of attention weights on specific words). Changing the location of the Eiffel Tower, for instance, should not lead the model to over-focus on the Eiffel Tower while neglecting other contexts (given that attention weights sum to 1). For this reason, imposing restrictions only when the attention weights surpass their normal levels is both necessary and beneficial, as this deviation can cause side effects like specificity failure (discussed in Section 3).
>
> For the slight drop in the generalization metric caused by the SADR method (less than 3%), it is important to note that prior evaluation systems often ignored specificity failure, achieving near 100% generalization at the cost of severe overfitting. What we aim to accomplish is to ensure that the knowledge retrieval process after editing remains as safe and close to the original Transformer’s knowledge extraction mechanism as possible. We also believe a stable knowledge editing method is more critical than achieving near 100% generalization accuracy.

---

> ### Author Response · Authors · 2024-11-21
> **Response to Reviewer RScL(Part II)**
>
> > - W2: Analysis on the reason for sepecificity failure.
>
> This is indeed a very meaningful and important question for our work. First, we would like to emphasize that **the specificity problem we investigate in this paper refers to cases where the model’s ability is negatively affected when content related to the edited knowledge appears in the context** (as noted in lines 44–46).
>
> Indeed, the edit vector's direction, space, and norm [1,2,3] can influence the model's specificity performance. However, the referenced works primarily focus on preserving general knowledge and capabilities, rather than addressing the specificity failure that arises when the edited subject appears in the context. To explore the relevance of these factors to the specificity failure problem studied in our work, we conducted a correlation analysis. Specifically, we compared four factors—**attention drift**, **hidden state norm post-editing**, **L2 distance between hidden states pre- and post-editing**, and the **cosine similarity of hidden states pre- and post-editing**—with the probability of $P(o_{\text{edit}})$ in specificity tasks.
>
> | **Factor**                  | **Pearson Coefficient (Distracting Neighborhood Task)** | **Pearson Coefficient (Relation Task)** |
> |-----------------------------|--------------------------------------------------------|----------------------------------------|
> | **Attention Drift**         | **0.49**                                               | **0.62**                              |
> | **Hidden State Norm**       | 0.01                                                   | 0.31                                   |
> | **L2 Distance (Hidden States)** | 0.01                                                   | 0.31                                   |
> | **Cosine Similarity (Hidden States)** | 0.02                                                   | -0.15                                  |
>
> The results show that, compared to the direction or norm of the edit vector, **attention drift has a more direct and significant impact on specificity failure.** We have also included this experiment in Appendix D.3 of the revised version.
>
> **Intuitively,** the attention mechanism is likely a key factor contributing to specificity failure. Previous studies have shown that when language models recall factual associations, the attention mechanism extracts answers from the hidden states of the subject[4,5]. Editing methods primarily modify the hidden states of the edited subject, which then influence the final output through the attention mechanism. In traditional editing methods (e.g., ROME discussed in Section 2.2), the optimization objective explicitly trains the model to predict the new $o_{\text{edit}}$ given $(s, r)$. **This may create a shortcut, where the subject’s hidden state is shaped in a way that makes it prone to being overly prioritized by the attention mechanism.** Consequently, whenever the edited subject appears, the model disproportionately outputs $o_{\text{edit}}$, satisfying the optimization objective while inadvertently causing specificity failure.
>
> **Experimentally,** as demonstrated in Section 3.3, we show that attention weights are a necessary condition for specificity failure. By **replacing only** **the post-editing attention weights with the pre-editing attention weights**—while keeping all other components unchanged (e.g., MLP outputs)—we observe a significant reduction in the probability of incorrect answers and a corresponding increase in the probability of correct ones during specificity tasks. This result strongly suggests that attention drift is **a primary driver and a necessary cause** of specificity failure.
>
>
>
> $Ref:$
>
> [1] AlphaEdit: Null-Space Constrained Knowledge Editing for Language Models
>
> [2] Knowledge Circuits in Pretrained Transformers
>
> [3] Perturbation-Restrained Sequential Model Editing
>
> [4] Dissecting Recall of Factual Associations in Auto-Regressive Language Models
>
> [5] Locating and Editing Factual Associations in GPT

---

> ### Author Response · Authors · 2024-11-21
> **Response to Reviewer RScL(Part III)**
>
> > - W3:  Comparison between the attention-based and decoding-based constraint methods.
>
> DeCK [1] primarily encourages the model to output words with greater differences from the original distribution by employing contrastive decoding, thereby increasing the model's confidence in the edited facts. However, DeCK will increase the model's confidence in its outputs, which leads to more severe specificity failure. Therefore, the formula needs to be modified to constrain the divergence between the model’s output and the original distribution. By modifying the Equation 6 in [1] as follows:
>
> $\mathcal F(\mathbb P_{\text{edited}}(x), \mathbb P_{\text{unedited}}(x)) = \log \mathbb P_{\text{edited}}(x) - 0.5 \log \left(\frac{\mathbb
>  P_{\text{edited}}(x)}{\mathbb P_{\text{unedited}}(x)}\right),$
>
> we can constrain the difference between the edited and unedited distributions during decoding. We refer to this approach as constrained decoding, and the results are as follows:
>
> | **Editor**                       | **ES** | **PS** | **NS** | **RS** | **DNS** |
> | -------------------------------- | ------ | ------ | ------ | ------ | ------- |
> | **None**                         | 20.86  | 17.70  | 82.43  | 79.73  | 61.99   |
> | **ROME**                         | 99.88  | 99.58  | 80.26  | 11.94  | 30.42   |
> | **+ contrastive decoding(DeCK)** | 100.00 | 99.94  | 26.12  | 0.39   | 11.94   |
> | **+ constrained decoding**       | 93.09  | 94.46  | 80.73  | 40.28  | 41.84   |
> | **+ SADR**                       | 99.76  | 96.36  | 80.86  | 27.75  | 49.32   |
>
> Although this approach significantly improves the metrics for the Relation task, its performance on all other tasks is decreased. Using decoding methods alone cannot fundamentally address the model's specificity failure and may also significantly harm the success rate of edits.
>
> $Ref:$
>
> [1] Decoding by Contrasting Knowledge: Enhancing LLMs’ Confidence on Edited Facts.

---

> ### Comment · Reviewer_RScL · 2024-11-25
> **Response to the Rebuttal**
>
> Thanks for the author's response, and they have dealt with some of my concerns, but I'm still not convinced of the generalization failure and I think this is the limitation.
> Anyway, the author's response makes sense; and it is a good work.
> I raised my score.

---

### Official Review · Reviewer_pEaf · 2024-11-03

**Soundness:** 4
**Presentation:** 4
**Contribution:** 3
**Rating:** 8
**Confidence:** 4

**Summary:**

The use of LLMs in real-world scenarios and applications creates the need for procedures to correct and update the knowledge in these models. The aim here is to change the model's knowledge without costly retraining in order to prevent hallucinations or correct obsolete facts without diminishing the model's performance.
Recently, the research field of knowledge editing has emerged, in which various techniques such as fine tuning, in-context editing, memory-based and locate-then-edit methods have already been proposed. The disadvantage of these methods is that they can negatively influence the model, especially if information of the edited knowledge triple or related content appears in the context. The study in this paper has set itself the task of shedding more light on this phenomenon, investigating its cause and proposing a method to prevent or mitigate this overcompensation of the edited model. In order to investigate the deteriorating specificity performance of an edited model, the authors develop two metrics and show that even a single updated fact can lead to a so-called specificity error.
An examination of these errors leads to the realization that they are  mainly caused by attention activations, the attention module places too  much focus on the edited information (attention drift) and ultimately  predicts an incorrect token. Consequently, the authors propose selective  attention drift restriction (SADR) as a method to mitigate this false focus.

**Strengths:**

The paper impresses with its consistently comprehensible and stringent argumentation. The authors start with a problem of a current methodology, prove that this problem exists, identify the underlying significant cause and can thus propose a solution method for the problem. The paper is comprehensibly written and error-free throughout, the illustrations and tables are helpful and well chosen. An additional plus is the ablation study, which deals with the trade-off between editing success and specificity.

**Weaknesses:**

A look at the appendix shows that the experiments for this article were much more extensive than stated in the actual paper. in addition to further details and results of the experiments described, further results for additional editing methods (WISE, MEND) and additional data sets can be found here. A human evaluation is also attached. It is a pity that even the section on limitations and future work did not find space in the main text. A minor weakness of the paper could be that it is not made clearer why the experiments are limited to locate-then-edit methods, although it is emphasized that the specificity error also occurs with meta-learning and parameter-preserving methods.
Typo line 47: Paris

**Questions:**

•	It is mentioned that there are specificity errors for models of all types. Have parameter preserving or meta-learning methods also been investigated? It might be interesting to know the RS/RM and DNS/DNM scores for methods like GRACE or ICE.
•	I would suggest adding at least the scores for MEMIT and PMET to Table 1

---

> ### Author Response · Authors · 2024-11-21
> **Response to Reviewer pEaf**
>
> We greatly appreciate your insightful feedback and constructive suggestions, and thank you for your recognition of our work. Here are our responses to your concerns:
>
> > - Weakness: Why the experiments are limited to locate-then-edit methods? Typo line 47: Paris
>
> Due to paper length constraints, we focuse our analysis of specificity failure and experiments with SADR on locate-then-edit methods to maintain consistency and clarity. We choose the locate-then-edit approach in the main text because it is a mainstream method in knowledge editing, offering state-of-the-art performance across many benchmarks with low computational demands. To illustrate the generalizability of the specificity issue and the proposed SADR method, we conduct extended experiments across more editing methods and datasets in the appendix.
>
> The typo issue has been corrected in the revised version of the paper.
>
> > - Questions: Have parameter preserving or meta-learning methods also been investigated? What's the RS/RM and DNS/DNM scores for methods like GRACE or ICE?  Adding the scores for MEMIT and PMET to Table 1.
>
> In Appendix E2, we provide metrics for WISE and MEND, which represent these two categories of methods. Our analysis shows that specificity failure remains a significant issue across these approaches, as evidenced by their RS and DNS scores compared to the original model.
>
> We have also evaluated the GRACE method on our tasks, with the results summarized below:
>
> | **Editor** | **ES** | **PS** | **NS** | **RS** |**RM** | **DNS** |**DNM** |
> | ---------- | ------ | ------ | ------ | ------ | ------ | ------- |------- |
> | **None**   | 20.86  | 17.70  | 82.43  | 79.73  | 8.83  | 61.99   |13.81  |
> | **GRACE**  | 100.00 | 31.00  | 59.78  | 29.83  | 4.00  | 60.33   |13.60   |
>
> The results indicate that GRACE does not perform well on the paraphrase task in the Counterfact editing benchmark.  For specificity tasks, GRACE shows some success in the Distracting Neighborhood scenario (likely due to its limited generalization capability) but still suffers from substantial overfitting in the Relation task. We acknowledge the importance of further exploring methods like ICE and plan to include them in future investigations.
>
> Additionally, to provide a more comprehensive illustration of specificity failure, we have added the scores for MEMIT and PMET to Table 1.

---

### Author Response · Authors · 2024-11-21
**General Response**

We sincerely thank all reviewers for their thoughtful and constructive feedback, as well as the time and effort in reviewing our work. We are delighted to see that all the reviewers recognize the importance of specificity failure as a critical issue in knowledge editing. Additionally, many reviewers acknowledge our analysis of specificity failure and affirm the generalizability and effectiveness of SADR.

For each reviewer’s concerns, we have provided detailed clarifications. We hope that our responses successfully address the remaining issues, and we are happy to answer any additional questions during the discussion phase.

We have also submitted a **revised version of the manuscript**, with the following updates:

1. Fixed typos and resolved missing references, as pointed out by reviewers DhFe and pEaf.
2. Added an efficiency analysis of SADR in the appendix.
3. Included additional correlation analyses in the appendix, investigating more factors related to specificity failure. The results further demonstrate that attention drift has a more direct impact on specificity failure compared to other factors.
4. Added a discussion on the reasons for attention drift in the appendix.

---

### Meta-Review · Area_Chair_XCjZ · 2024-12-20

**Metareview:**

Previous knowledge editing approaches can negatively impact the model, especially when the edited knowledge or related content reappears in the context. This paper aims to shed light on this phenomenon, exploring its causes and proposing a method to prevent or reduce this overcompensation in the edited model. To investigate the decline in specificity performance of an edited model, the authors develop two metrics and demonstrate that even a single updated fact can cause a specificity error. An analysis of these errors reveals that they are primarily driven by attention activations—specifically, the attention module overfocuses on the edited information (attention drift), leading to incorrect predictions. To address this issue, the authors introduce Selective Attention Drift Restriction (SADR) as a method to mitigate false focus. All reviewers agree that this paper makes a clear contribution to the field. It is recommended that the authors carefully revise the paper according to the reviewers' suggestions.

**Additional Comments On Reviewer Discussion:**

All reviewers believe that the paper makes a clear contribution.

---

### Decision · Program_Chairs · 2025-01-22

Accept (Poster)